# Defending against Backdoor Attacks via Module Switching

**Weijun Li**[1]   **Ansh Arora**[2]   **Xuanli He**[3]   **Mark Dras**[1]   **Qiongkai Xu**[1]*

[1]Macquarie University, Sydney, NSW, Australia
[2]University of Massachusetts Amherst, Amherst, MA, USA
[3]University College London, London, UK
`weijun.li1@hdr.mq.edu.au`  `qiongkai.xu@mq.edu.au`

## Abstract

Backdoor attacks pose a serious threat to deep neural networks (DNNs), allowing adversaries to implant triggers for hidden behaviors in inference. Defending against such vulnerabilities is especially difficult in the post-training setting, since end-users lack training data or prior knowledge of the attacks. Model merging offers a cost-effective defense; however, latest methods like weight averaging (WAG) provide reasonable protection when multiple homologous models are available, but are less effective with fewer models and place heavy demands on defenders. We propose a module-switching defense (MSD) for disrupting backdoor shortcuts. We first validate its theoretical rationale and empirical effectiveness on two-layer networks, showing its capability of achieving higher backdoor divergence than WAG, and preserving utility. For deep models, we evaluate MSD on Transformer and CNN architectures and design an evolutionary algorithm to optimize fusion strategies with selective mechanisms to identify the most effective combinations. Experiments shown that MSD achieves stronger defense with fewer models in practical settings, and even under an underexplored case of collusive attacks among multiple models–where some models share same backdoors–switching strategies by MSD deliver superior robustness against diverse attacks. Code is available at https://github.com/weijun-l/module-switching-defense.

## 1 Introduction

Backdoor attacks pose a particularly insidious threat to modern neural networks. By injecting crafted triggers into a small portion of training data (Gu et al., 2017; Chen et al., 2017), an adversary trains models to behave normally on clean inputs yet exhibit malicious behavior when triggers appear. The combination of stealth and effectiveness makes them a critical security concern, particularly as training increasingly relies on large-scale, uncurated web data (Halfacree, 2025).

This threat is amplified by the shift toward a "post-training" paradigm, where practitioners adopt models without visibility into their origins. This trend manifests in several prominent scenarios: (1) open-source model platforms, *e.g.,* HuggingFace (Wolf et al., 2019), which facilitate widespread reuse and finetuning of pretrained models; (2) multi-expert systems like Mixture-of-Experts (MoE), where a router dynamically selects among specialized models trained on heterogeneous data (Fedus et al., 2022; Zhou et al., 2022); (3) one-shot Federated Learning (Guha et al., 2019; Dai et al., 2024), which allows a central server aggregating models from distributed clients once. While these trends accelerate innovation, they also share a vulnerability: the opacity of training data and processes provides fertile ground for adversarial attacks (Huynh & Hardouin, 2023).

The post-training paradigm presents a dual challenge: its opacity not only enables hidden backdoors but also undermines traditional defenses. Many existing defenses assume access to training-time resources, such as the original data for filtering (He et al., 2023; 2024), a trusted auxiliary dataset for fine-tuning (Liu et al., 2018; Zhang et al., 2022; Min et al., 2023; Zhao et al., 2024), or the optimization procedure for trigger inversion (Tao et al., 2022; Sur et al., 2023). Without these

---
*Corresponding author.

resource hypotheses, model merging (Izmailov et al., 2018; Matena & Raffel, 2022; Aristimuño, 2026)-originally designed for knowledge aggregation-emerges as a compelling defense strategy that leverages multi-model availability and suppresses backdoors (Arora et al., 2024).

Nonetheless, model merging is not a panacea. Existing approaches face three main constraints: (1) methods such as WAG (Arora et al., 2024) and DAM (Yang et al., 2025) typically require 3 to 6 homologous models to achieve effective backdoor suppression, which imposes a heavy burden on defenders; (2) strategies guided by trusted criteria, curated data, or proxy models (Yang et al., 2025; Chen et al., 2024) depend on resources that are often scarce in untrusted environments; and (3) although compromised auxiliary models can be used as defensive references (Li et al., 2024; Tong et al., 2024), they may introduce additional risks (He et al., 2025).

To overcome these constraints, we propose *Module Switching*, a defense framework that selectively exchanges network modules across models from related tasks and domains. The key insight is that backdoors operate as learned "shortcuts", exploiting spurious correlations to trigger malicious behavior (Gardner et al., 2021; He et al., 2023; Ye et al., 2024; Li et al., 2025). These shortcuts are typically localized within specific modules, yet different backdoored models rarely implant them in the same location. Swapping corresponding modules thus disrupts these fragile pathways, replacing compromised components with benign counterparts and thereby neutralizing the vulnerability.

This mechanism brings two key benefits: (1) compared to weight averaging, it blocks backdoor transmission with fewer models, providing a more practical defense; and (2) it also offers robustness in an underexplored case of collusive attacks, where some models share backdoors and weight averaging degrades to fewer-model performance, whereas module switching remains effective. We empirically demonstrate both benefits in Section 5.2.

We formulate shortcut disruption as an optimization problem: searching for module-switching strategies that break shortcut connections within a given model architecture. By combining heuristic scoring and an evolutionary algorithm, we obtain an index table that specifies which source model should fill each

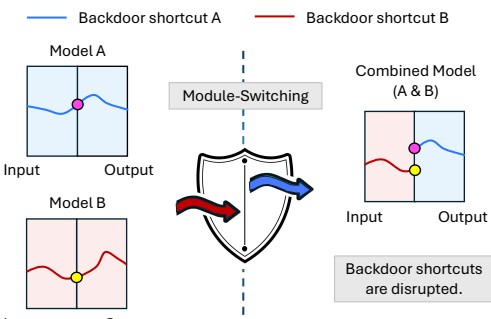

Figure 1: An illustration of Module-Switching Defense (MSD). By switching weight modules between compromised models (*left*), the spurious correlations (shortcuts) learned from backdoored tasks are disrupted in the combined model (*right*).

module slot. As this scheme relies solely on architectural information, it generalizes well across tasks and is transferable to models sharing the same structure (*e.g.,* one strategy applicable to *RoBERTa* (Liu et al., 2019b) can be reused for *DeBERTa* (He et al., 2021)).

Our **Module-Switching Defense (MSD)** applies the strategy by assigning each module across the network a source-model index and recombining the selected modules to construct candidate models. Then, we identify the most robust candidate by comparing their representations on a small clean validation set (requiring only 20–50 samples per class and no poisoned data). Since MSD is structure-driven, it is task-agnostic, counters a wide spectrum of backdoor threats, and preserves utility for downstream tasks. Our key contributions are as follows.

- We propose and develop MSD, which (1) establishes heuristic rules (Section 4.2) to guide evolutionary search for module-switching strategies (Section 4.3), and (2) defines a feature-distance criterion to select the best candidate combination (Section 4.4).

- We conduct study on shallow networks to analyze and interpret the mechanism of module-switching on backdoor mitigation and semantic preservation (Section 3).

- We empirically validate favorable properties of MSD, including (1) stronger defense under more practical, fewer-model constraints, and (2) robustness against the underexplored collusive attack surface where multiple models share the same backdoors (Section 5.2).

## 2 RELATED WORK

**Backdoor Attacks.** Backdoor attacks implant hidden vulnerabilities in DNNs, activating only when specific triggers appear in the input while behave normal on benign data. They can be broadly

categorized by implanting methods: (1) *Data-poisoning attacks* inject trigger patterns into a small portion of the datasets with manipulated labels to train compromised models. Since being first discovered by Gu et al. (2017), these attacks have evolved with diverse trigger designs in both vision (Nguyen & Tran, 2021; Li et al., 2021b; Xu et al., 2023b; Huynh et al., 2024) and text domains (Dai et al., 2019; Kurita et al., 2020; Qi et al., 2021b;c). In contrast, (2) *Weight-poisoning attacks* directly modify model weights to embed backdoors (Dumford & Scheirer, 2020; Kurita et al., 2020). Backdoor attacks can be considered as correlating trigger patterns with predefined predictions in DNNs, activated in inference (Gardner et al., 2021; He et al., 2023). Our work focuses on defending against *data-poisoning attacks*, given their widespread adoption and potential risks.

**Backdoor Defense.** Backdoor defenses can be classified by deployment stage into (1) *training-phase* and (2) *test-phase* methods. *Training-phase* defenses treat poisoned data as outliers, aiming to detect and removing them based on distinctive activation or learning patterns (Li et al., 2021a; He et al., 2023; 2024). *Test-phase* defenses operate on inputs or the model itself: data-level approaches reverse-engineer triggers (Wang et al., 2019; 2023) or detect poisoned inputs (Qi et al., 2021a; Gao et al., 2024; Xie et al., 2024; Hou et al., 2024), while model-level strategies detect trojaned models (Liu et al., 2019a; Wang et al., 2020; 2024a; Su et al., 2024) or purify models through pruning, fine-tuning, or other adaptations (Wu & Wang, 2021; Liu et al., 2018; Zhang et al., 2022; Xu et al., 2023a; Zhao et al., 2024; Cheng et al., 2024) or unlearning (Wu & Wang, 2021; Zeng et al., 2022; Li et al., 2023a).

Detection-based methods aim to identify poisoned samples or compromised models and therefore address a complementary defense dimension relative to model-level repair techniques. While traditional model purification demands proxy data and retraining, recent research has focused on model combination strategies requiring fewer assumptions and lower computational costs (Arora et al., 2024; Yang et al., 2025; Chen et al., 2024; Li et al., 2024; Tong et al., 2024). Building on this line of work, we propose a model fusion approach that reduces dependency on trusted resources while mitigating threats by disrupting spurious correlations in constituent models.

## 3 MODULE SWITCHING IN TWO-LAYER NEURAL NETWORKS

We theoretically and empirically examine whether *module switching* in two-layer networks disrupts backdoor patterns introduced during fine-tuning while preserving pretrained semantics. We find that swapping layer weights deviates more from backdoor patterns than weight averaging (WAG) (Arora et al., 2024; Wang et al., 2024b), yielding improved robustness against backdoored inputs.

### 3.1 PRELIMINARY SETUP

**Setup and Notation.** We consider two-layer networks defined as $f(\boldsymbol{x}; \theta) = \boldsymbol{W}_2\, \sigma(\boldsymbol{W}_1 \boldsymbol{x})$, with input $\boldsymbol{x} \in \mathbb{R}^N$ and parameters $\theta := \{\boldsymbol{W}_1, \boldsymbol{W}_2\}$, and activation function $\sigma(\cdot)$ (linear or non-linear). Training progresses in two stages: a *pretraining* stage, where shared weights $\boldsymbol{W}_1 \in \mathbb{R}^{K \times N}$ and $\boldsymbol{W}_2 \in \mathbb{R}^{N \times K}$ learn general semantics, followed by a *fine-tuning* stage that introduces updates ($\Delta \boldsymbol{W}_1^*$ and $\Delta \boldsymbol{W}_2^*$) to encode backdoor behavior in individual models $\mathcal{M}^*$.

In a linear network with identity activation, the fine-tuned model is $\mathcal{M}(\boldsymbol{x}) = (\boldsymbol{W}_2 + \Delta \boldsymbol{W}_2^*)(\boldsymbol{W}_1 + \Delta \boldsymbol{W}_1^*)\boldsymbol{x}$, which expands to a semantic term $\boldsymbol{S} = \boldsymbol{W}_2 \boldsymbol{W}_1$ and a backdoor component

$$\boldsymbol{B}^* = \boldsymbol{W}_2 \Delta \boldsymbol{W}_1^* + \Delta \boldsymbol{W}_2^* \boldsymbol{W}_1 + \epsilon^*, \tag{1}$$

such that $\mathcal{M}^*(\boldsymbol{x}) = (\boldsymbol{S} + \boldsymbol{B}^*)\boldsymbol{x}$, where $\epsilon^* = \Delta \boldsymbol{W}_2^* \Delta \boldsymbol{W}_1^*$ represents a second-order interaction. The $\epsilon$-terms are typically much smaller in magnitude than first-order terms (*i.e.,* $\boldsymbol{W}_2 \Delta \boldsymbol{W}_1^* + \Delta \boldsymbol{W}_2^* \boldsymbol{W}_1$). We empirically verify this in Appendix C, and accordingly omit the $\epsilon$-term in subsequent analysis.

### 3.2 THEORETICAL ANALYSIS

We first define the weight averaged and the module switched models, together with the notion of output distances between these combinations and their constituent models. These distances will be used to quantify how WAG and the switched models differ from the constituent backdoor models.

**Definition 1** (Weight-Averaged Model). *Let $i$ and $j$ index two fine-tuned backdoor models. Averaging the weights of $\mathcal{M}^i$ and $\mathcal{M}^j$ defines the* Weight-Averaged (WAG) model, *with parameters:*

$$\theta^{\mathrm{wag}} := \left\{ \frac{1}{2}\left(\boldsymbol{W}_1 + \Delta \boldsymbol{W}_1^i\right) + \frac{1}{2}\left(\boldsymbol{W}_1 + \Delta \boldsymbol{W}_1^j\right), \frac{1}{2}\left(\boldsymbol{W}_2 + \Delta \boldsymbol{W}_2^i\right) + \frac{1}{2}\left(\boldsymbol{W}_2 + \Delta \boldsymbol{W}_2^j\right) \right\}.$$

*Assuming a linear network as above, we decompose the model as $\mathcal{M}^{\text{wag}}(\boldsymbol{x}) = (\boldsymbol{S} + \boldsymbol{B}^{\text{wag}})\,\boldsymbol{x}$, where $\boldsymbol{S}$ denotes the shared pretrained semantics, and the backdoor component is equivalent to*

$$\boldsymbol{B}^{\text{wag}} = \frac{1}{2}\boldsymbol{W}_2\left(\Delta\boldsymbol{W}_1^i + \Delta\boldsymbol{W}_1^j\right) + \frac{1}{2}\left(\Delta\boldsymbol{W}_2^i + \Delta\boldsymbol{W}_2^j\right)\boldsymbol{W}_1.$$

**Definition 2** (Distance between Outputs from WAG and Constituent Models). *Under identity activation, $\ell_2$ distances between the WAG model and the two constituent models $\mathcal{M}^i$ and $\mathcal{M}^j$ are:*

$$\|\mathcal{D}^{\text{wag},i}\| = \|\mathcal{M}^{\text{wag}}(\boldsymbol{x}) - \mathcal{M}^i(\boldsymbol{x})\| = \frac{1}{2}\|\left(\boldsymbol{W}_2(\Delta\boldsymbol{W}_1^j - \Delta\boldsymbol{W}_1^i) + (\Delta\boldsymbol{W}_2^j - \Delta\boldsymbol{W}_2^i)\boldsymbol{W}_1\right)\boldsymbol{x}\|,$$

$$\|\mathcal{D}^{\text{wag},j}\| = \|\mathcal{M}^{\text{wag}}(\boldsymbol{x}) - \mathcal{M}^j(\boldsymbol{x})\| = \frac{1}{2}\|\left(\boldsymbol{W}_2(\Delta\boldsymbol{W}_1^i - \Delta\boldsymbol{W}_1^j) + (\Delta\boldsymbol{W}_2^i - \Delta\boldsymbol{W}_2^j)\boldsymbol{W}_1\right)\boldsymbol{x}\|.$$

**Definition 3** (Module-Switched Models). *Swapping one layer between $\mathcal{M}^i$ and $\mathcal{M}^j$ yields two possible switched models, each with its own parameters, semantic-backdoor decomposition:*

$$\theta^{ij} := \{\boldsymbol{W}_1 + \Delta\boldsymbol{W}_1^i,\ \boldsymbol{W}_2 + \Delta\boldsymbol{W}_2^j\}, \quad \mathcal{M}^{ij}(\boldsymbol{x}) = (\boldsymbol{S} + \boldsymbol{B}^{ij})\boldsymbol{x}, \quad \boldsymbol{B}^{ij} = \boldsymbol{W}_2\Delta\boldsymbol{W}_1^i + \Delta\boldsymbol{W}_2^j\boldsymbol{W}_1,$$

$$\theta^{ji} := \{\boldsymbol{W}_1 + \Delta\boldsymbol{W}_1^j,\ \boldsymbol{W}_2 + \Delta\boldsymbol{W}_2^i\}, \quad \mathcal{M}^{ji}(\boldsymbol{x}) = (\boldsymbol{S} + \boldsymbol{B}^{ji})\boldsymbol{x}, \quad \boldsymbol{B}^{ji} = \boldsymbol{W}_2\Delta\boldsymbol{W}_1^j + \Delta\boldsymbol{W}_2^i\boldsymbol{W}_1.$$

**Definition 4** (Distance between Outputs from Switched and Constituent Models). *Under identity activation, $\ell_2$ distances between the switched model $\mathcal{M}^{ij}$ and the two constituent models are:*

$$\|\mathcal{D}^{ij,i}\| = \|\mathcal{M}^{ij}(\boldsymbol{x}) - \mathcal{M}^i(\boldsymbol{x})\| = \|(\Delta\boldsymbol{W}_2^j - \Delta\boldsymbol{W}_2^i)\boldsymbol{W}_1\boldsymbol{x}\|,$$

$$\|\mathcal{D}^{ij,j}\| = \|\mathcal{M}^{ij}(\boldsymbol{x}) - \mathcal{M}^j(\boldsymbol{x})\| = \|\boldsymbol{W}_2(\Delta\boldsymbol{W}_1^i - \Delta\boldsymbol{W}_1^j)\boldsymbol{x}\|.$$

*The analogous results for $\|\mathcal{D}^{ji,i}\|$ and $\|\mathcal{D}^{ji,j}\|$ hold with swapped indices (see Equation (5)).*

To show the improved divergence achieved by module switching, we next compare how far the switched models move relative to the constituent backdoor models, in contrast to WAG.

**Theorem 1** (Module Switching Exceeds WAG in Backdoor Divergence). *Under identity activation, the total backdoor divergence of the Weight-Averaged (WAG) model is upper bounded by the average divergence of the switched models:*

$$\|\mathcal{D}^{\text{wag},i}\| + \|\mathcal{D}^{\text{wag},j}\| \leq \frac{1}{2}\left(\|\mathcal{D}^{ij,i}\| + \|\mathcal{D}^{ij,j}\| + \|\mathcal{D}^{ji,i}\| + \|\mathcal{D}^{ji,j}\|\right). \tag{2}$$

This theorem confirms the rationale that module switching on average yields stronger suppression of backdoor-specific patterns than weight averaging.

**Proposition 1** (The Existence of a More Divergent Switched Model). *Given Theorem 1, there exists at least one switched model with greater backdoor divergence than Weight-Averaged (WAG) model:*

$$\|\mathcal{D}^{\text{wag},i}\| + \|\mathcal{D}^{\text{wag},j}\| \leq \max\left\{\|\mathcal{D}^{ij,i}\| + \|\mathcal{D}^{ij,j}\|,\ \|\mathcal{D}^{ji,i}\| + \|\mathcal{D}^{ji,j}\|\right\}. \tag{3}$$

This proposition shows that the least backdoor-aligned switched model exceeds the WAG model in backdoor divergence, underscoring the importance of selecting the least aligned candidate and motivating the selection step in Section 4.4. Appendix D details proofs of Theorem 1 and Proposition 1.

**Utility Loss.** Having established the divergence properties of backdoor components, we next examine whether module switching compromises utility. For a model $\mathcal{M}^*$, we measure its utility loss as the distance between its outputs to benign semantics, *i.e.,* $L^*(\boldsymbol{x}) := \mathcal{M}^*(\boldsymbol{x}) - S\boldsymbol{x} = B^*\boldsymbol{x}$. The switched models satisfy the identity $L^{ij} + L^{ji} = L^i + L^j$ (see Appendix E), implying that the total loss of a switched pair is equivalent to the sum of its constituents. To assess individual models, we empirically measure each switched model's loss relative to its originals and find that the relative utility loss remains low (see Appendix F), demonstrating promising utility preservation.

## 3.3 EMPIRICAL ANALYSIS

We simulate 1000 linear and non-linear two-layer networks, each *pretrained* on a shared semantic component $\boldsymbol{S} \sim \mathcal{N}(\boldsymbol{0}, 1)$ and *fine-tuned* with a backdoor component $\boldsymbol{B}^* \sim \mathcal{N}(\boldsymbol{0}, 0.1^2)$. For each fine-tuned pair $\mathcal{M}^i$ and $\mathcal{M}^j$, we construct the corresponding WAG model $\mathcal{M}^{\text{wag}}$ and switched models $\mathcal{M}^{ij}$ and $\mathcal{M}^{ji}$. We evaluate output alignment with (1) the semantic direction $\boldsymbol{S}\boldsymbol{x}$, measured by $d_S = \|\text{norm}(f(\boldsymbol{x};\theta)) - \text{norm}(\boldsymbol{S}\boldsymbol{x})\|$; and (2) the backdoor direction $\boldsymbol{B}^*\boldsymbol{x}$, measured by $d_B = \|\text{norm}(f(\boldsymbol{x};\theta) - \boldsymbol{S}\boldsymbol{x}) - \text{norm}(\boldsymbol{B}^*\boldsymbol{x})\|$, where $\text{norm}(\boldsymbol{v}) = \boldsymbol{v}/\|\boldsymbol{v}\|$.

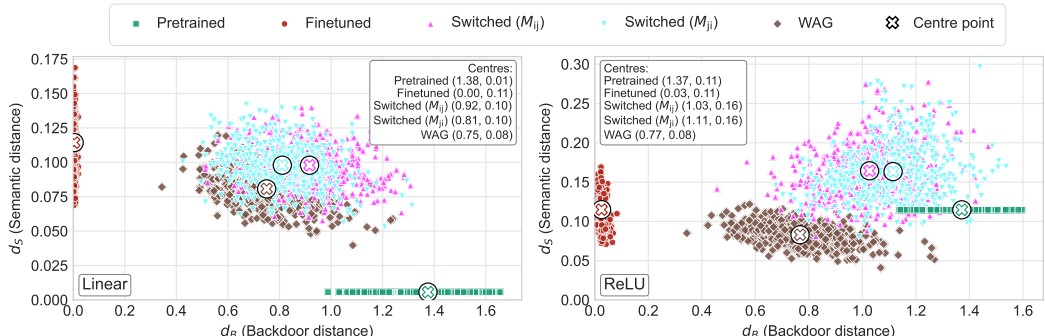

Figure 2: Euclidean distances between normalized output vectors of *pretrained*, *fine-tuned*, *WAG*, and *switched* two-layer networks, relative to the semantic direction $\boldsymbol{Sx}$ and the backdoor directions $\boldsymbol{B^*x}$, under linear (left) and ReLU (right) activations.

Figure 2 presents 2D scatter plots comparing output distances across all model types under both linear and ReLU (Nair & Hinton, 2010; Agarap, 2018) activations. More results with various activations are provided in Appendix G. We observe that while *fine-tuned* models stay close to their respective backdoor patterns $\boldsymbol{B^*}$, the *WAG* model shifts farther away, and the *switched* models diverge even more, indicating stronger backdoor suppression. All models remain near the semantic term $\boldsymbol{S}$, confirming preserved functionality.

## 4 MODULE SWITCHING DEFENSE

In this section, we extend the findings on module switching to more complicated deep neural networks and develop a comprehensive defense pipeline. We begin by introducing the problem setting in Section 4.1, followed by establishing a set of heuristic rules to guide the search for effective module switching strategies in Section 4.2. Next, we adapt an evolutionary algorithm for searching the optimal strategy in Section 4.3, guiding switched models construction and selection in Section 4.4.

### 4.1 PRELIMINARIES

**Threat Model.** We study data poisoning attacks where an attacker modifies a subset of a clean dataset $\mathcal{D}_c = \{(x_c, y_c)\}$ into poisoned samples $\mathcal{D}_p = \{(x_p = g_t(x_c), y_p)\}$ using a trigger function $g_t$ and target label $y_p$. The poisoned data is used to train a backdoored model or shared with others for training, resulting in trojaned models being widely available via model-sharing platforms.

**Defender Capability.** The defender downloads potentially trojaned models and aims to purify them before deployment. They have white-box access and a small clean validation set (20–50 samples per class), but no knowledge of the triggers or poisoned data. They can access multiple (as few as two) domain-relevant models of uncertain integrity and may combine them using the validation set.

**Neural Network Architecture.** We adopt Transformer models (Vaswani et al., 2017), chosen for their strong performance and popularity in both text and vision. Each model has $L$ layers, composed of a self-attention block and a feed-forward network (FFN), both followed by residual connections (He et al., 2016). We abbreviate the six core modules (the attention block's query ($W_q$), key ($W_k$), value ($W_v$), output ($W_o$), and the FFN's input ($W_i$) and output ($W_p$)) as $\{Q, K, V, O, I, P\}$.

To assess cross-modality applicability, we also examine vision architectures, including *Vision Transformers (ViT)* (Wu et al., 2020) and convolutional networks (CNNs) (Lecun et al., 1998; Krizhevsky et al., 2012). For ViT models, we apply the same module abstraction used for text-based Transformers. For CNNs such as the ResNet family (He et al., 2016), each convolution-batch normalization weight pair is treated as a module (*e.g.,* the first conv-bn pair in each BasicBlock denoted as $C1$).

### 4.2 SCORING RULES FOR MODULE SWITCHING

In Section 3, we studied weight switching in two-layer networks, where replacing weights disrupts spurious correlations, eliminating undesired patterns while preserving utility. Extending to deep models, we hypothesize that breaking backdoor propagation paths can similarly deactivate them.

Given the structural complexity of deep networks, we define heuristic rules to guide the search for module combinations that disrupt backdoor paths in both feedforward and residual streams (Elhage

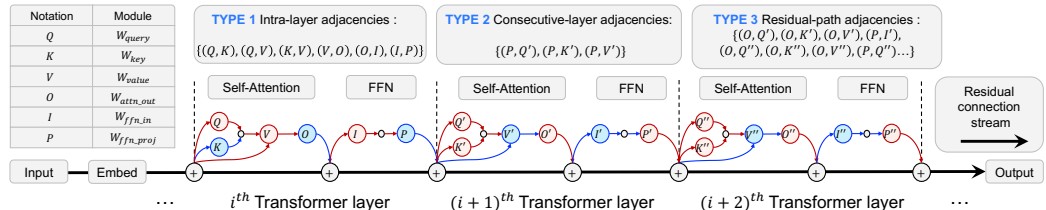

Figure 3: The fused model combines modules from different models (shown as red and blue nodes), considering three types of module adjacency in Transformers, as illustrated at the top.

et al., 2021). We identify three types of adjacency that may support poison transmission (illustrated in Figure 3): (1) intra-layer (within the same layer), (2) consecutive-layer (adjacent layers), and (3) residual (via skip connections). Additionally, we introduce a (4) balance penalty to avoid overusing any single model and a (5) diversity reward to encourage varied combinations across layers.

These rules serve as criteria for computing an overall score of a given module-switching strategy, evaluating how well it adheres to the principles. Detailed rules and types are in Appendix H.1.

## 4.3 EVOLUTIONARY MODULE SWITCHING SEARCH

We frame the search for effective module-switching strategies as a discrete Neural Architecture Search (NAS) problem (White et al., 2023). Let $\mathcal{S}$ denote the space of switching strategies, where each $s \in \mathcal{S}$ assigns a source model index to each module: $s : \{1, \ldots, L\} \times M \to \{1, \ldots, N\}, M = \{Q, K, V, O, I, P\}$, where $L$ is the number of layers and $N$ the number of source models.

**Fitness Evaluation.** Each strategy $s$ is scored by

$$F(s) = -\lambda_1 A_{\text{intra}}(s) - \lambda_2 A_{\text{cons}}(s) - \lambda_3 A_{\text{res}}(s) - \lambda_4 B_{\text{bal}}(s) + \lambda_5 R_{\text{div}}(s), \qquad (4)$$

where $A_{\text{intra}}$, $A_{\text{cons}}$, and $A_{\text{res}}$ penalize adjacency violations (Section 4.2), $B_{\text{bal}}$ penalizes module imbalance, and $R_{\text{div}}$ rewards diversity. By default, we set all $\lambda_k$ to 1.0. Higher $F(s)$ indicates stronger disruption of potential backdoor paths. Details of each term are in Appendix H.2.

**Search Algorithm.** As the scores by $F(s)$ is non-differentiable over a large discrete space, we adopt evolutionary search (Miller et al., 1989), well-suited to optimizing implicit objectives (Zhou et al., 2021). We adopt the aging regularized evolution algorithm (Real et al., 2019), modifying it in two key ways: (1) fitness is computed directly using the heuristic scoring function $F$, without model training or validation; and (2) low-scoring strategies are discarded, replacing aging regularization (So et al., 2019). As outlined in Algorithm 1, it evolves a population through tournament selection (line 11), mutation (line 12), and fitness-based dropping (line 13). Appendix K presents example searched strategies.

---

**Algorithm 1** Evolutionary Module-Switching Search

1: **Input:** population $P$, generations $G$, children per iter $C$, number of models $N$, layers $L$, module set $M$.
2: $population \leftarrow \varnothing$
3: $gen\_count \leftarrow 0$
4: **while** $|population| < P$ **do**
5: $\quad indiv.strategy \leftarrow \text{RANDOMSTRATEGY}(N, L, M)$
6: $\quad indiv.fitness \leftarrow \text{CALCSCORE}(indiv.strategy)$
7: $\quad population.append(indiv)$
8: **end while**
9: **while** $gen\_count < G$ **do**
10: $\quad$ **for** $i \leftarrow 1$ **to** $C$ **do**
11: $\quad\quad parent \leftarrow \text{TOURNAMENTSELECT}(population)$
12: $\quad\quad child.strategy \leftarrow \text{MUTATION}(parent)$
13: $\quad\quad child.fitness \leftarrow \text{CALCSCORE}(child.strategy)$
14: $\quad\quad population.append(child)$
15: $\quad$ **end for**
16: $\quad sort(population)$ ▷ by descending $fitness$ score
17: $\quad population \leftarrow population[0 : P]$ ▷ truncate to P
18: $\quad gen\_count \leftarrow gen\_count + 1$
19: **end while**
20: **Output:** $\text{BESTSTRATEGY} \leftarrow population[0].strategy$

---

## 4.4 SWITCHED MODELS CONSTRUCTION AND SELECTION

The searched strategy $T$ can be used to switch modules among a group of victim models $\mathcal{M} = \{\mathcal{M}_1, \ldots, \mathcal{M}_N\}$ to fuse a *candidate pool*, which on average exceeds the WAG model in backdoor

divergence (as Theorem 1) and guarantees the existence of at least one candidate with higher divergence (as Proposition 1). This motivates us to develop a feature-distance-based method to select the least-backdoor-aligned candidate from the pool.

**Suspect-class Detection.** We first use the final-layer embedding of [CLS] token to detect the suspect class, based on the insight that backdoored models prioritize trigger features (Fu et al., 2023; Yi et al., 2024; Wang et al., 2024a). For each $m \in \mathcal{M} \cup \{\text{WAG}(\mathcal{M})\}$ and class $c$, we optimize a random input to induce prediction of $c$, yielding a dummy final-layer [CLS] feature $z_{m,c}^{\text{dum}}$. Its average cosine distance to clean features over a few non-$c$ samples is accumulated across models: $S(c) = \sum_m \text{avg}\big[1 - \cos(z_{m,c}^{\text{dum}}, z_{m,\neg c}^{\text{clean}})\big]$. The class with the highest score, $c^* = \arg\max_c S(c)$, is deemed suspicious, and the corresponding WAG dummy feature $z^* = z_{\text{WAG},c^*}^{\text{dum}}$ is used as a fixed reference. For CNNs, the same procedure is applied using the global average pooled feature of the final convolutional layer in place of the [CLS] embedding.

---

**Algorithm 2** Switched Model Selection

1: **Input:** Victim models $\mathcal{M} = \{\mathcal{M}_1, \ldots, \mathcal{M}_N\}$; clean set $\mathcal{D}_c$; switching strategy $T$.
2: $wag \leftarrow \text{WAG}(\mathcal{M})$      ▷ weight averaging over $\mathcal{M}$
3: $models \leftarrow \mathcal{M} \cup \{wag\}$
4: $score \leftarrow \text{ZEROVECTOR}(\texttt{num\_classes})$
5: **for** $m \in models$ **do**
6:     **for** $c \in$ candidate classes **do**
7:         $x_{\text{dummy}} \leftarrow \text{OPTIMIZEINPUT}(m, x_{\text{random}}, c)$
8:         $z_{\text{dummy}} \leftarrow \text{FORWARD}(m, x_{\text{dummy}})$
9:         $z_{\text{clean}} \leftarrow \text{FORWARD}(m, \mathcal{D}_c, \texttt{non-}c)$
10:        $score[c] \mathrel{+}= \text{MEANCOSINEDIST}(z_{\text{dummy}}, z_{\text{clean}})$
11:        $\text{DUMMYFEATURE}[m][c] \leftarrow z_{\text{dummy}}$
12:     **end for**
13: **end for**
14: $c^* \leftarrow \arg\max_c score[c]$    ▷ suspect target class
15: $z^* \leftarrow \text{DUMMYFEATURE}[wag][c^*]$
16: $candidates \leftarrow \text{MODULESWITCH}(T, \mathcal{M})$
17: **for** $m \in candidates$ **do**
18:     $z \leftarrow \text{FORWARD}(m, \mathcal{D}_c, \texttt{non-}c^*)$
19:     $m.dist \leftarrow \text{MEANCOSINEDIST}(z, z^*)$
20: **end for**
21: **Output:** $\arg\max_m m.dist$

---

**Candidate Selection.** Applying $T$ to $\mathcal{M}$ gives candidates $m \in \mathcal{C}(T, \mathcal{M})$ (*e.g.,* $\mathcal{M}^{ij}$ and $\mathcal{M}^{ji}$). Each $m$ is scored by $d(m) = \text{avg}\big[1 - \cos(z^*, f_m(\boldsymbol{x}))\big]$, the mean cosine distance between its [CLS] features on a few clean, non-$c^*$ samples $\boldsymbol{x}$ and the WAG dummy $z^*$. The winner $m^* = \arg\max_{m \in \mathcal{C}(T, \mathcal{M})} d(m)$ is the one least aligned with backdoor features and, by Proposition 1, has better defense than WAG. The pipeline detailed in Algorithm 2, avoids exhaustive trojan detection process (Wang et al., 2019; 2020; 2024a; Su et al., 2024), yet reliably selects robust candidates.

## 5 EXPERIMENTS

### 5.1 EXPERIMENTAL SETUP

**Datasets.** We primarily evaluate our method in the text domain using three NLP datasets: **SST-2** (Socher et al., 2013), **MNLI** (Williams et al., 2018), and **AG News** (Zhang et al., 2015). Following previous work (Arora et al., 2024), we apply a poison rate of 20% in training, and additionally test settings with 10% and 1%. To further assess cross-domain applicability, we evaluate on two vision datasets: **CIFAR-10** (Krizhevsky et al., 2009) and **TinyImageNet** (Le & Yang, 2015), where a poison rate of 5% is used. The statistics of the used datasets are reported in Table 6 (Appendix I.1). The poisoned test sets, used solely for evaluation, are generated by adding triggers to validation samples outside the target class, while the defenders are restricted to access only the clean test set.

**Backdoor Attacks.** We evaluate our defense against four text-based backdoor attacks that poison data by modifying and relabeling clean samples. Two are insertion-based: **BadNet** (Kurita et al., 2020), which adds rare-word triggers {*"cf", "mn", "bb", "tq", "mb"*}, and **InsertSent** (Dai et al., 2019), which inserts trigger phrases {*"I watched this movie", "no cross, no crown"*}. The other two are stealthier: Learnable Word Substitution (**LWS**) (Qi et al., 2021c), which uses synonym substitution, and Hidden-Killer (**Hidden**) (Qi et al., 2021b), which applies syntactic paraphrasing.

For the vision domain, we use attacks that inject digital patterns, such as **BadNet** (Gu et al., 2017) and **BATT** (Xu et al., 2023b), as well as stealthier methods like the warping-based **WaNet** (Nguyen & Tran, 2021) and the object-based **PhysicalBA** (Li et al., 2021b). To further challenge our defense, we also evaluate it against the **Adaptive-Patch** attack (Qi et al., 2023). All poisoned vision datasets and models are generated using the **BackdoorBox** toolkit (Li et al., 2023b).

Table 1: Performance comparison across backdoor attacks on **SST-2** using *RoBERTa-large*. Best results are in blue. * indicates results averaged over four variants; same for subsequent tables.

| Defense | CACC | Attack Success Rate (ASR)↓ | | | | | Defense | CACC | Attack Success Rate (ASR)↓ | | | | |
|---|---|---|---|---|---|---|---|---|---|---|---|---|---|
| | | BadNet | Insert | LWS | Hidden | AVG. | | | BadNet | Insert | LWS | Hidden | AVG. |
| Benign | 95.9 | 4.1 | 2.2 | 12.8 | 16.5 | 8.9 | Z-Def | 95.6* | 4.6 | 1.8 | 97.3 | 35.7 | 34.9 |
| Victim | 95.9* | 100.0 | 100.0 | 98.0 | 96.5 | 98.6 | ONION | 92.8* | 56.8 | 99.9 | 85.7 | 92.9 | 83.8 |
| *Combined: BadNet + InsertSent* | | | | | | | *Combined: BadNet + HiddenKiller* | | | | | | |
| WAG | 96.3 | 56.3 | 7.4 | - | - | 31.9 | WAG | 96.1 | 63.9 | - | - | 29.0 | 46.4 |
| TIES | 95.9 | 88.7 | 17.0 | - | - | 52.9 | TIES | 96.0 | 90.4 | - | - | 36.9 | 63.6 |
| DARE | 96.5 | 57.8 | 36.3 | - | - | 47.1 | DARE | 96.7 | 36.5 | - | - | 47.6 | 41.9 |
| Ours | 96.2 | 36.9 | 7.1 | - | - | 22.0 | Ours | 96.1 | 40.5 | - | - | 27.7 | 34.1 |
| *Combined: BadNet + LWS* | | | | | | | *Combined: Benign + BadNet* | | | | | | |
| WAG | 96.2 | 74.0 | - | 50.3 | - | 62.2 | WAG | 96.1 | 39.3 | - | - | - | 39.3 |
| TIES | 95.9 | 88.1 | - | 66.1 | - | 77.1 | TIES | 95.7 | 69.2 | - | - | - | 69.2 |
| DARE | 96.2 | 60.4 | - | 62.5 | - | 61.4 | DARE | 96.4 | 43.2 | - | - | - | 43.2 |
| Ours | 96.0 | 41.7 | - | 39.0 | - | 40.4 | Ours | 96.1 | 12.2 | - | - | - | 12.2 |

**Defense Baselines.** We compare against seven defenses across text and vision: three model-merging approaches applicable to both domains–**TIES** (Yadav et al., 2023), **DARE** (Yu et al., 2024), and **WAG** (Arora et al., 2024)–and two domain-specific data purification methods per modality. In text, **Z-Def.** (He et al., 2023) and **ONION** (Qi et al., 2021a) perform outlier detection; in vision, **Cut-Mix** (Yun et al., 2019) disrupts triggers via patch mixing, and **ShrinkPad** (Li et al., 2021b) reduces vulnerability by shrinking and padding inputs. All baselines use open-source implementations with default settings (see Appendix I.3 for details).

**Evaluation Metrics.** We assess utility and defense with Clean Accuracy (**CACC**) and Attack Success Rate (**ASR**) (Qi et al., 2021a;c; Arora et al., 2024). CACC is the accuracy on clean samples, with higher values indicating better utility. ASR is the accuracy on a poisoned test set, where all samples are attacked and relabeled to the target class; higher ASR indicates greater vulnerability.

**Implementation Details.** We use *RoBERTa-large* (Liu et al., 2019b), *BERT-large* (Devlin et al., 2019), and *DeBERTa-large* (He et al., 2021) for text experiments; and *ViT* (Wu et al., 2020) as well as pretrained *ResNet-18* and *ResNet-50* (He et al., 2016; TorchVision maintainers and contributors, 2016) for vision experiments. NLP models are fine-tuned for 3 epochs with Adam (Kingma & Ba, 2015) at $2 \times 10^{-5}$, and vision models for 10 epochs with SGD (Bottou, 2010) at $1 \times 10^{-2}$.

We evaluate two-model merging in both domains and additionally consider multi-model merging for text. All experiments are run with three random seeds on a single Nvidia A100 GPU, and results are averaged. The evolutionary search runs for 2 million generations on a single Intel Core i9-14900K CPU, taking 2.6 hours for two models and 4.3 hours for four models. Since the strategy is structure-driven and task-agnostic, only one search is required per architecture. For model selection, discussed in Section 4.4, we use 50 samples per class to choose candidate models and ablate this to 20 in Section 5.3. Selection takes less than a minute on both SST-2 and CIFAR-10.

## 5.2 MAIN RESULTS

**Mitigation of Textual Backdoor Attacks.** We evaluate our defense with *RoBERTa-large* on **SST-2**, **MNLI**, and **AG News**. Partial SST-2 results appear in Table 1, with full results in Appendix J.1. We evaluate merging backdoored models to examine robustness against attacks, and merging backdoored with benign models to examine resistance to backdoor transfer. A unified strategy from our evolutionary algorithm (see Figure 7) is applied consistently across all cases.

Across all datasets and model pairs, our method shows strong defense while preserving clean accuracy. Merging BadNet and InsertSent yields an ASR of 22.0%, compared to 31.9% for WAG. With BadNet and LWS (a stealthier attack), it reaches 40.4%, over 21.0% lower than baselines (typically above 60%). These results demonstrate that even with compromised models, our approach disrupts spurious correlations and mitigates backdoors.

When merging a benign model with compromised ones, our method consistently yields low ASRs across four combinations. In the BadNet-controlled case, it achieves 12.2%, 27.1% better than WAG. This indicates that our method blocks unintended backdoor effects, unlike approaches that preserve utility but risk new vulnerabilities. While Z-Def performs well against insertion-based attacks (with training data access), it is less effective against attacks with subtle trigger patterns.

Table 2: Performance comparison across backdoor attacks on the **CIFAR-10** dataset using *ViT*.

| Defense | CACC | BadNet | WaNet | BATT | PBA | AVG. | Defense | CACC | BadNet | WaNet | BATT | PBA | AVG. |
|---|---|---|---|---|---|---|---|---|---|---|---|---|---|
| Benign | 98.8 | 10.1 | 10.2 | 7.7 | 10.1 | 9.5 | CutMix | 97.7* | 87.1 | 70.6 | 99.9 | 64.9 | 80.6 |
| Victim | 98.5* | 96.3 | 84.7 | 99.9 | 89.4 | 92.6 | ShrinkPad | 97.3* | 14.4 | 51.3 | 99.9 | 88.3 | 63.5 |
| *Combined: BadNet + WaNet* | | | | | | | *Combined: BadNet + BATT* | | | | | | |
| WAG | 98.7 | 13.7 | 10.6 | - | - | 12.2 | WAG | 98.9 | 10.1 | - | 42.9 | - | 26.5 |
| TIES | 98.6 | 11.9 | 10.7 | - | - | 11.3 | TIES | 98.9 | 10.1 | - | 47.9 | - | 29.0 |
| DARE | 98.8 | 83.3 | 10.2 | - | - | 46.7 | DARE | 99.0 | 69.2 | - | 26.8 | - | 48.0 |
| Ours | 98.7 | 12.3 | 10.5 | - | - | 11.4 | Ours | 98.7 | 10.2 | - | 32.6 | - | 21.4 |
| *Combined: BadNet + PhysicalBA* | | | | | | | *Combined: Benign + PhysicalBA* | | | | | | |
| WAG | 99.0 | 39.6 | - | - | 39.5 | 39.6 | WAG | 99.0 | - | - | - | 10.1 | 10.1 |
| TIES | 99.0 | 38.9 | - | - | 38.9 | 38.9 | TIES | 98.8 | - | - | - | 10.2 | 10.2 |
| DARE | 99.0 | 72.2 | - | - | 72.2 | 72.2 | DARE | 99.9 | - | - | - | 10.1 | 10.1 |
| Ours | 98.7 | 18.5 | - | - | 18.4 | 18.5 | Ours | 98.9 | - | - | - | 10.1 | 10.1 |

**Mitigation of Vision Backdoor Attacks.** We assess our method on the **CIFAR-10** and **TinyImageNet** datasets using a 12-layer *ViT* (Wu et al., 2020) model. Partial results for CIFAR-10 are shown in Table 2, with full results presented in Appendix J.2. The evolutionary search yields the module-switching strategy in Figure 14, applied across all vision experiments.

Our method consistently defends against all attack combinations while preserving utility. For example, in the BadNet + PhysicalBA case, it lowers ASR to 18.5%, outperforming all baselines by at least 20.4%. These results demonstrate the robustness of our strategy in disrupting spurious correlations and its effectiveness across domains with different input characteristics.

**Three-Model Fusion Defense.** When three backdoored models are available, even baseline WAG already shows strong results. However, our module-switching approach achieves consistently stronger defense. Using the strategy in Figure 15 (see Appendix J.3), MSD reduces the average ASR to below 20% across different combinations, outperforming WAG as reported in Table 12.

**Merging Models with Collusive Backdoors.** Although WAG achieves relatively low ASRs when combining multiple models, in a realistic yet underexplored scenario some models may share identical backdoors. In such settings, WAG degenerates to fewer-model behavior, reducing its defensive effectiveness. In contrast, our module-switching strategy is more resilient, as it strategically disrupts these recurring shortcuts. Using the strategy in Figure 16 (see Appendix J.3), MSD outperforms WAG under collusion, as shown in Table 13, demonstrating robustness against collusive models.

**Comparison of Different Strategies.** We compare two evolutionary search strategies–with and without early stopping–shown in Figures 7 and 8, and report their fitness scores in Table 14 of Appendix J.4. The early stopping terminates the search when no improvement in fitness score is observed over 100,000 iterations. We observe a positive correlation between the fitness score and defense performance: the adopted strategy without early stopping achieves a higher score and reduces the ASR by 27.2%. Based on score breakdowns and visualizations, we attribute the improvement to fewer residual rule violations, which more effectively disrupt subtle spurious correlations.

**Diversity of Discovered Strategies.** We further examine the structural diversity of strategies produced by the evolutionary search. Using three strategies obtained from different random seeds (Figures 7 and 9), we compute their module-level overlap. As detailed in Appendix J.5, only 10 out of 144 module positions coincide across all strategies (6.94%), with no region or module type exhibiting higher consistency than others. This demonstrates that MSD does not rely on a narrow set of critical layers but instead induces broad structural disruption, which helps mitigate backdoor effects and makes the searched strategies transferable and reusable across different scenarios.

**Candidate Selection Results.** Our method generates multiple asymmetric module allocation candidates, with selection guided by the process in Section 4.4. While the selected candidate consistently performs well, we also analyze the unselected ones (see Table 15 in Appendix J.6). In most cases, our method correctly identifies the top-performing candidate, outperforming other options by a significant margin. Even when an unselected candidate achieves a lower ASR in specific cases, our chosen candidate remains competitive with both the best alternative and the WAG baseline.

## 5.3 ABLATION STUDIES

**Importance of Heuristic Rules.** We ablate each of the first three rules from Section 4.2 to evaluate their individual contributions. As shown in Table 16 (Appendix J.7), removing any rule typically

degrades performance, highlighting the complementary effect of the full rule set. Visualizations in Figures 11 to 13 show that each ablation yields distinct strategy patterns.

**Generalization across Architectures.** We apply our method to *RoBERTa-large*, *BERT-large*, and *DeBERTa-v3-large* under three settings. As shown in Table 17 (Appendix J.8), our approach consistently outperforms WAG across all tests. Importantly, we reuse the same searched strategy from Figure 7, demonstrating strong cross-model generalization and supporting practical scalability.

To further evaluate generality beyond Transformer families, we also extend MSD to CNN architectures, including ResNet-18 and ResNet-50 on CIFAR-10. The searched strategies for these models are shown in Appendix K (Figures 17 and 18), and the full quantitative results are presented in Appendix J.8 (Table 18). Across diverse combinations, MSD achieves comparable or superior ASR reduction relative to WAG while maintaining similar clean accuracy. These results demonstrate that MSD naturally transfers to CNN-based models, reinforcing its cross-domain robustness.

**Minimum Clean Data Requirement.** We examine the impact of reducing clean supervision from 50 to 20 samples per class on SST-2 across three architectures. Results in Table 17 (Appendix J.9) show our method still selects low-ASR candidates, suggesting effectiveness with limited clean data.

**Performance under Varying Poisoning Rates.** We test robustness under 20%, 10%, and 1% poisoning rates on SST-2 using *RoBERTa-large*. As shown in Table 19 (Appendix J.10), our method consistently achieves lower ASR than WAG across different attacks and poisoning levels.

**Robustness to Adaptive Attacks.** We consider two types of threat scenarios: attacks that are adaptive to MSD and challenging backdoor patterns that introduce stronger shortcut behaviors. First, we consider an attacker who knows the deployed module selection strategy and retrains only those modules on poisoned data. Our approach counters this by generating diverse strategies using different random seeds. Even if one strategy (Figure 7) is compromised, alternatives (Figure 9) remain effective, as demonstrated on SST-2 with *RoBERTa-large* (Table 20). Second, we evaluate a challenging backdoor pattern, Adaptive-Patch (Qi et al., 2023), which is not MSD-specific but induces more complex shortcut behavior. Using a transferability-based strategy (Figure 14), our method consistently demonstrates strong defensive performance (Table 21). A detailed analysis of both scenarios is provided in Appendix J.11.

**Robustness to Label-Inconsistent and Identical Backdoors.** A practical consideration for model merging is that the obtained models may be trained by different attackers targeting different labels, or they may encode identical backdoor triggers. We therefore examine two challenging settings: (1) models with inconsistent target labels, and (2) models trained with the same backdoor trigger. In the first case, where each model has a different target label, our method maintains strong defensive performance. In the second case, where models share the same trigger, our method again substantially reduces ASR compared to WAG, as shown in Table 22 (Appendix J.12).

**Efficiency Analysis.** We compare the computational efficiency of MSD with representative baselines in the two-model setting. The comparison is summarized in Table 3.

MSD requires a one-time architecture-dependent search of 2.6 hours that can be performed offline, after which the merging step takes only 16 seconds. In contrast, deployment-time search methods such as DARE need to rerun a greedy search for every new model pair, taking

Table 3: Efficiency comparison.

| Phase | DARE | TIES | WAG | MSD |
|---|---|---|---|---|
| Search | 2.5 hrs | – | – | 2.6 hrs |
| Merge | – | 1 min | 10 s | 16 s |

approximately 2.5 hours per deployment. Since the MSD strategy can be reused for all models that share the same architecture, the amortized deployment cost becomes negligible, providing a practical efficiency advantage while maintaining strong defensive performance.

## 6 CONCLUSION

In this paper, we propose Module-Switching Defense (MSD), a post-training backdoor defense that disrupts shortcuts of spurious correlations by strategically switching weight modules between (compromised) models. MSD does not rely on trusted reference models or training data and remains effective with a couple of models. Using heuristic rules and evolutionary search, we establish a transferable module fusion strategy that mitigates various backdoor attacks while preserving their task utility. Empirical results on text and vision tasks confirm its outstanding defense performance, and strong generalization capability, highlighting its practicality in real-world applications.

ACKNOWLEDGMENTS

We would like to express our appreciation to the anonymous reviewers for their valuable feedback. This research was undertaken with the assistance of resources from the National Computational Infrastructure (NCI Australia), an NCRIS enabled capability supported by the Australian Government. We also express our gratitude to SoC Incentive Fund, FSE strategic startup grant and HDR Research Support Funding for supporting both travel and research.

ETHICS STATEMENT

This paper presents an efficient post-training defense against backdoor attacks on machine learning models. By strategically combining model weight modules from either clean or compromised models, our approach disrupts backdoor propagation while preserving model utility. We demonstrated the usage of MSD to strengthen the security of machine learning models in both natural language processing and computer vision. All models and datasets used in this study are sourced from established open-source platforms. The discovered MSD templates will be released to facilitate further research on defense study. While we do not anticipate any direct negative societal consequences, we hope this work encourages further research into more robust defense mechanisms.

REPRODUCIBILITY STATEMENT

We describe our method in detail in Section 4, with two key algorithms presented in Algorithm 1 and Algorithm 2. Experimental settings are documented in Section 5.1, and the searched outputs of the algorithms are included in Appendix K. To support reproducibility, we release our implementation code at: https://github.com/weijun-l/module-switching-defense.

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

## A    LIMITATIONS

While our study demonstrates the effectiveness of Module-Switching Defense (MSD) across a range of classification tasks in NLP and CV, our current scope is limited to classification-based settings. Backdoor attacks in generative models operate through notably different mechanisms, and extending MSD to such scenarios remains an important direction for future research.

## B    GENERATIVE LLM USAGE STATEMENT

We used ChatGPT and Gemini for surface-level edits, such as grammar checks, phrasing refinement, and table caption formatting to improve readability.

## C    EMPIRICAL VALIDATION OF THE SECOND-ORDER INTERACTION MAGNITUDE

We empirically validate the condition adopted in Section 3, where the second-order interaction term $\epsilon = \Delta W_2 \Delta W_1$ is omitted due to its negligible magnitude relative to the first-order terms. This validation proceeds from three perspectives.

First, Figure 4 compares the Frobenius norms of the semantic term $S = W_2 W_1$, the first-order adaptation term $B = W_2 \Delta W_1 + \Delta W_2 W_1$, and the second-order residual $\epsilon = \Delta W_2 \Delta W_1$ across five derived networks. The left subfigure confirms that $\|\epsilon\|$ is consistently two orders of magnitude smaller than $\|S\|$ and well below $4\%$ of $\|B\|$. The right subfigure further reveals that the element-wise values of $\epsilon$ concentrate tightly around zero, contrasting with the heavier tails of $B$ and $S$.

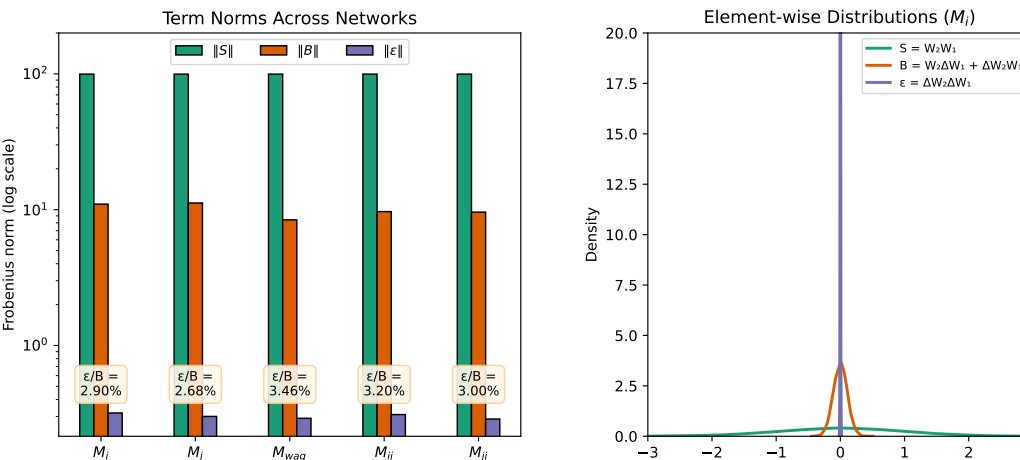

Figure 4: Frobenius norm and element-wise distribution of the semantic, first-order, and second-order terms across five network configurations. While the first-order term dominates the residual behavior, the second-order interaction $\epsilon = \Delta W_2 \Delta W_1$ remains negligible in both scale and distribution.

Second, Table 4 reports $\|\epsilon\| / \|B\|$ ratios across five network variants under varying backdoor strengths, where perturbations are sampled from zero-mean Gaussian noise with increasing variance. The inclusion of error bars (mean $\pm$ standard deviation) reflects variation across multiple runs. In typical scenarios where the backdoor signal is weak or comparable to the main semantic component, the second-order interaction consistently remains below $4\%$ of the first-order term. Even under exaggerated settings where the backdoor signal is scaled to $1.5\times$ or $2\times$ the semantic strength, $\|\epsilon\| / \|B\|$ remains within a stable range of $5\%$–$7\%$, reaffirming the negligible and bounded nature of second-order interactions across regimes.

Additionally, we extend this analysis to deep transformer-based (Vaswani et al., 2017) models by computing $\|\epsilon\| / \|B\|$ for the attention weight product, where $W_1$ and $W_2$ denote the key ($K$) and

Table 4: Relative magnitude of second-order interactions, reported as $\|\varepsilon\|/\|\boldsymbol{B}\|$, across networks and backdoor strengths. All models are evaluated with $\boldsymbol{S} \sim \mathcal{N}(\boldsymbol{0}, 1)$ and perturbations $\boldsymbol{B} \sim \mathcal{N}(\boldsymbol{0}, \sigma^2)$.

| Semantic Dist | Backdoor Dist | $\|\varepsilon\|/\|\boldsymbol{B}\|$ for Different Shallow Models | | | | |
|---|---|---|---|---|---|---|
| | | $\mathcal{M}^i$ | $\mathcal{M}^j$ | $\mathcal{M}^{wag}$ | $\mathcal{M}^{ij}$ | $\mathcal{M}^{ji}$ |
| | $\boldsymbol{B} \sim \mathcal{N}(\boldsymbol{0}, 0.1^2)$ | 2.82±0.19% | 2.70±0.20% | 3.42±0.23% | 2.95±0.15% | 3.14±0.21% |
| | $\boldsymbol{B} \sim \mathcal{N}(\boldsymbol{0}, 0.5^2)$ | 1.98±0.21% | 1.90±0.28% | 1.78±0.18% | 1.75±0.12% | 1.76±0.18% |
| $\boldsymbol{S} \sim \mathcal{N}(\boldsymbol{0}, 1.0^2)$ | $\boldsymbol{B} \sim \mathcal{N}(\boldsymbol{0}, 1.0^2)$ | 3.24±0.22% | 3.10±0.32% | 2.33±0.17% | 2.33±0.12% | 2.30±0.17% |
| | $\boldsymbol{B} \sim \mathcal{N}(\boldsymbol{0}, 1.5^2)$ | 4.77±0.25% | 4.48±0.33% | 3.18±0.14% | 3.09±0.15% | 2.97±0.12% |
| | $\boldsymbol{B} \sim \mathcal{N}(\boldsymbol{0}, 2.0^2)$ | 6.31±0.27% | 6.28±0.35% | 4.14±0.29% | 4.06±0.14% | 3.92±0.18% |

query ($Q$) projection matrices, respectively, and $QK^\top := \boldsymbol{W}_2\boldsymbol{W}_1$. The weight changes $\Delta\boldsymbol{W}_1$, $\Delta\boldsymbol{W}_2$ are computed relative to the original pretrained *RoBERTa-large* (Liu et al., 2019b) weights. All models are trained on **SST-2** (Socher et al., 2013), including both benign and backdoored variants such as **BadNet** (Kurita et al., 2020), **InsertSent** (Dai et al., 2019), learnable word substitution (**LWS**) (Qi et al., 2021c), and Hidden-Killer (**Hidden**) (Qi et al., 2021b).

As shown in Table 5, across all pairwise combinations of these models, the relative magnitude of second-order interactions consistently remains below $4\%$. Each reported value reflects the mean and standard deviation computed across all 24 layers of *RoBERTa-large*. This pattern holds across both original and recombined variants ($\mathcal{M}^{\text{wag}}$, $\mathcal{M}^{ij}$, $\mathcal{M}^{ji}$), confirming the stability of second-order contributions in practical transformer settings.

Table 5: Relative magnitude of second-order interactions, reported as $\|\varepsilon\|/\|\boldsymbol{B}\|$, computed from the key ($K$) and query ($Q$) projection matrices in *RoBERTa-large* models trained on **SST-2**.

| Combination | $\|\varepsilon\|/\|\boldsymbol{B}\|$ for Attention Weight Product ($QK^T$) in *RoBERTa-large* Models | | | | |
|---|---|---|---|---|---|
| ($\mathcal{M}^i + \mathcal{M}^j$) | $\mathcal{M}^i$ | $\mathcal{M}^j$ | $\mathcal{M}^{wag}$ | $\mathcal{M}^{ij}$ | $\mathcal{M}^{ji}$ |
| *BadNet + InsertSent* | 3.53±0.77% | 3.24±0.61% | 2.43±0.39% | 2.68±0.51% | 2.61±0.50% |
| *BadNet + LWS* | 3.53±0.77% | 3.30±0.65% | 2.46±0.4% | 2.71±0.46% | 2.68±0.49% |
| *BadNet + Hidden* | 3.53±0.77% | 3.30±0.61% | 2.49±0.43% | 2.77±0.45% | 2.72±0.45% |
| *BadNet + Benign* | 3.53±0.77% | 3.27±0.58% | 2.52±0.42% | 2.78±0.47% | 2.73±0.48% |

Accordingly, we omit the second-order term $\epsilon$ in our definitions and proofs throughout the paper without loss of generality.

## D  PROOFS OF THEOREM 1 AND PROPOSITION 1

**Theorem 1** (Module Switching Exceeds WAG in Backdoor Divergence). *Under identity activation, the total backdoor divergence of the Weight-Averaged (WAG) model is upper bounded by the average divergence of the switched models:*

$$\|\mathcal{D}^{\mathrm{wag},i}\| + \|\mathcal{D}^{\mathrm{wag},j}\| \leq \frac{1}{2}\left(\|\mathcal{D}^{ij,i}\| + \|\mathcal{D}^{ij,j}\| + \|\mathcal{D}^{ji,i}\| + \|\mathcal{D}^{ji,j}\|\right). \tag{2}$$

**Proposition 1** (The Existence of a More Divergent Switched Model). *Given Theorem 1, there exists at least one switched model with greater backdoor divergence than Weight-Averaged (WAG) model:*

$$\|\mathcal{D}^{\mathrm{wag},i}\| + \|\mathcal{D}^{\mathrm{wag},j}\| \leq \max\left\{\|\mathcal{D}^{ij,i}\| + \|\mathcal{D}^{ij,j}\|,\ \|\mathcal{D}^{ji,i}\| + \|\mathcal{D}^{ji,j}\|\right\}. \tag{3}$$

*Proof.* From Definition 2 and 4, we have the following expressions for the backdoor divergences:

$$\begin{aligned}
\|\mathcal{D}^{\mathrm{wag},i}\| &= \frac{1}{2}\left\|\left(\boldsymbol{W}_2(\Delta\boldsymbol{W}_1^j - \Delta\boldsymbol{W}_1^i) + (\Delta\boldsymbol{W}_2^j - \Delta\boldsymbol{W}_2^i)\boldsymbol{W}_1\right)\boldsymbol{x}\right\|, \\
\|\mathcal{D}^{\mathrm{wag},j}\| &= \frac{1}{2}\left\|\left(\boldsymbol{W}_2(\Delta\boldsymbol{W}_1^i - \Delta\boldsymbol{W}_1^j) + (\Delta\boldsymbol{W}_2^i - \Delta\boldsymbol{W}_2^j)\boldsymbol{W}_1\right)\boldsymbol{x}\right\|, \\
\|\mathcal{D}^{ij,i}\| &= \left\|(\Delta\boldsymbol{W}_2^j - \Delta\boldsymbol{W}_2^i)\boldsymbol{W}_1\boldsymbol{x}\right\|, \quad \|\mathcal{D}^{ij,j}\| = \left\|\boldsymbol{W}_2(\Delta\boldsymbol{W}_1^i - \Delta\boldsymbol{W}_1^j)\boldsymbol{x}\right\|, \\
\|\mathcal{D}^{ji,i}\| &= \left\|\boldsymbol{W}_2(\Delta\boldsymbol{W}_1^j - \Delta\boldsymbol{W}_1^i)\boldsymbol{x}\right\|, \quad \|\mathcal{D}^{ji,j}\| = \left\|(\Delta\boldsymbol{W}_2^i - \Delta\boldsymbol{W}_2^j)\boldsymbol{W}_1\boldsymbol{x}\right\|.
\end{aligned} \tag{5}$$

**Linear relationships.**   By regrouping terms in the above definitions, we obtain the following vector identities:

$$\mathcal{D}^{\mathrm{wag},i} = \frac{1}{2}(\mathcal{D}^{ij,i} + \mathcal{D}^{ji,i}), \qquad \mathcal{D}^{\mathrm{wag},j} = \frac{1}{2}(\mathcal{D}^{ij,j} + \mathcal{D}^{ji,j}). \tag{6}$$

**Bounding the average switched model backdoor divergence.**   Substituting equation 6 into the norms and applying the triangle inequality (Tversky & Gati, 1982), we have:

$$\|\mathcal{D}^{\mathrm{wag},i}\| = \|\frac{1}{2}(\mathcal{D}^{ij,i} + \mathcal{D}^{ji,i})\| \leq \frac{1}{2}\left(\|\mathcal{D}^{ij,i}\| + \|\mathcal{D}^{ji,i}\|\right), \tag{7}$$

$$\|\mathcal{D}^{\mathrm{wag},j}\| = \|\frac{1}{2}(\mathcal{D}^{ij,j} + \mathcal{D}^{ji,j})\| \leq \frac{1}{2}\left(\|\mathcal{D}^{ij,j}\| + \|\mathcal{D}^{ji,j}\|\right). \tag{8}$$

Summing both inequalities gives:

$$\|\mathcal{D}^{\mathrm{wag},i}\| + \|\mathcal{D}^{\mathrm{wag},j}\| \leq \frac{1}{2}\left(\|\mathcal{D}^{ij,i}\| + \|\mathcal{D}^{ji,i}\| + \|\mathcal{D}^{ij,j}\| + \|\mathcal{D}^{ji,j}\|\right), \tag{9}$$

which proves Theorem 1.

**Bounding the maximum switched model backdoor divergence.**   Let:

$$C_1 := \|\mathcal{D}^{ij,i}\| + \|\mathcal{D}^{ij,j}\|, \qquad C_2 := \|\mathcal{D}^{ji,i}\| + \|\mathcal{D}^{ji,j}\|, \qquad G := \max\{C_1, C_2\}. \tag{10}$$

Since $C_1 + C_2 \leq 2G$, it follows that:

$$\|\mathcal{D}^{\mathrm{wag},i}\| + \|\mathcal{D}^{\mathrm{wag},j}\| \leq \frac{1}{2}(C_1 + C_2) \leq \max\{C_1, C_2\}, \tag{11}$$

which proves Proposition 1.  □

## E  DERIVATION OF UTILITY LOSS IDENTITY

As discussed in Section 3, utility loss in a two-layer network can be expressed as the difference from the benign semantic output. For a model $\mathcal{M}^*$, we define

$$L^*(\boldsymbol{x}) := \mathcal{M}^*(\boldsymbol{x}) - S\boldsymbol{x} = B^*\boldsymbol{x}. \tag{12}$$

According to the notation in Section 3 and the construction in Definition 3, the utility losses of constituent and switched models are

$$L^i(\boldsymbol{x}) = \big(W_2\Delta W_1^i + \Delta W_2^i W_1\big)\boldsymbol{x}; \; L^j(\boldsymbol{x}) = \big(W_2\Delta W_1^j + \Delta W_2^j W_1\big)\boldsymbol{x};$$
$$L^{ij}(\boldsymbol{x}) = \big(W_2\Delta W_1^i + \Delta W_2^j W_1\big)\boldsymbol{x}; \; L^{ji}(\boldsymbol{x}) = \big(W_2\Delta W_1^j + \Delta W_2^i W_1\big)\boldsymbol{x}. \tag{13}$$

Regrouping these expressions yields the key identity,

$$\begin{aligned} L^{ij}(\boldsymbol{x}) + L^{ji}(\boldsymbol{x}) &= \big(W_2\Delta W_1^i + \Delta W_2^j W_1\big)\boldsymbol{x} + \big(W_2\Delta W_1^j + \Delta W_2^i W_1\big)\boldsymbol{x} \\ &= \big(W_2\Delta W_1^i + \Delta W_2^i W_1\big)\boldsymbol{x} + \big(W_2\Delta W_1^j + \Delta W_2^j W_1\big)\boldsymbol{x} \\ &= L^i(\boldsymbol{x}) + L^j(\boldsymbol{x}). \end{aligned} \tag{14}$$

## F  EMPIRICAL EVALUATION OF UTILITY LOSS

As derived in Appendix E, the identity, $L^{ij}(\boldsymbol{x}) + L^{ji}(\boldsymbol{x}) = L^i(\boldsymbol{x}) + L^j(\boldsymbol{x})$, characterizes the combined loss of a switched pair relative to its constituent models. We now evaluate loss at the level of individual switched models with respect to the benign semantic output.

For any model $\mathcal{M}^*$, define its utility loss ratio relative to benign semantic $S$ as

$$r^*(\boldsymbol{x}) := \frac{\|L^*(\boldsymbol{x})\|}{\|S\boldsymbol{x}\|} \quad \text{for inputs with } \|S\boldsymbol{x}\| > 0. \tag{15}$$

To measure how a switched model compares to its originals, we denote

$$\begin{aligned} e^{ij,i}(\boldsymbol{x}) &:= r^{ij}(\boldsymbol{x}) - r^i(\boldsymbol{x}), \quad e^{ij,j}(\boldsymbol{x}) := r^{ij}(\boldsymbol{x}) - r^j(\boldsymbol{x}), \\ e^{ji,i}(\boldsymbol{x}) &:= r^{ji}(\boldsymbol{x}) - r^i(\boldsymbol{x}), \quad e^{ji,j}(\boldsymbol{x}) := r^{ji}(\boldsymbol{x}) - r^j(\boldsymbol{x}). \end{aligned} \tag{16}$$

Let $\mathcal{E} = \{e^{ij,i}, e^{ij,j}, e^{ji,i}, e^{ji,j}\}$ denote the collection of all signed differences between the switched models and their origins. A value close to zero indicates that the switched models $\mathcal{M}^{ij}$ and $\mathcal{M}^{ji}$ preserve the benign utility at a level comparable to their original models $\mathcal{M}^i$ and $\mathcal{M}^j$. Negative value further implies that a switched model provides representations *closer* to the benign semantics than those by corresponding original models.

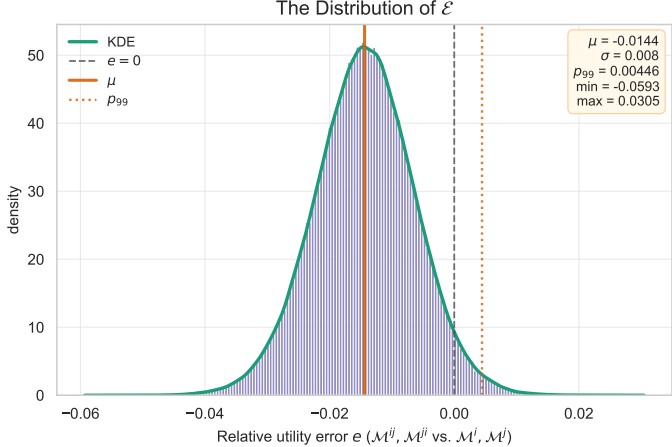

Figure 5: Empirical distribution of $\mathcal{E}$, the relative utility loss differences between switched models $(\mathcal{M}^{ij}, \mathcal{M}^{ji})$ and their originals $(\mathcal{M}^i, \mathcal{M}^j)$. Non-positive values indicate that a switched model is closer to the benign utility, with more negative values corresponding to closer alignment.

Figure 5 shows the empirical distribution of $\mathcal{E}$ aggregated over 1,000 randomly sampled model pairs $(\mathcal{M}^i, \mathcal{M}^j)$. The combined results yield a mean of $-0.014 \pm 0.008$, 99th percentile of 0.004, and a maximum of 0.031, suggesting that the maximum deviation from the originals is below 3%, while on average the switched models $\mathcal{M}^{ij}$ and $\mathcal{M}^{ji}$ incur slightly smaller loss relative to the benign utility. These findings demonstrate that utility is effectively preserved under module switching, compared to the original constituent models.

## G  MODULE SWITCHING WITH ADDITIONAL ACTIVATION FUNCTIONS

We extend the experiments from Section 3 to two additional activation functions: *tanh* and *sigmoid* (Dubey et al., 2022), in addition to the *linear* and *ReLU* results discussed in the main text. For each activation, we simulate 1000 pairs of *fine-tuned* models $\mathcal{M}^i$ and $\mathcal{M}^j$ with a shared pretrained semantic component $\boldsymbol{S} \sim \mathcal{N}(\boldsymbol{0}, 1^2)$ and individual backdoor shifts $\boldsymbol{B}^* \sim \mathcal{N}(\boldsymbol{0}, 0.1^2)$. We then construct the weight-averaged model $\mathcal{M}^{\text{wag}}$ and the module-switched models $\mathcal{M}^{ij}$ and $\mathcal{M}^{ji}$, as defined in Definitions 1 and 3.

Figure 6 visualizes the semantic and backdoor alignment of each model type across the four activation functions. Consistently across activations, we observe that:

- *Fine-tuned* models remain closely aligned with their respective backdoor direction $\boldsymbol{B}^*\boldsymbol{x}$;
- *WAG* models deviate more from the backdoor pattern;
- *Switched* models exhibit the larger distance to backdoor patterns, indicating stronger mitigation;
- All model types maintain proximity to the semantic output $\boldsymbol{S}\boldsymbol{x}$, confirming that semantic information is preserved.

These results generalize the findings in Figure 2 to a broader range of nonlinear activations, reinforcing the conclusion that module switching more effectively disrupts backdoor behavior while retaining semantic utility.

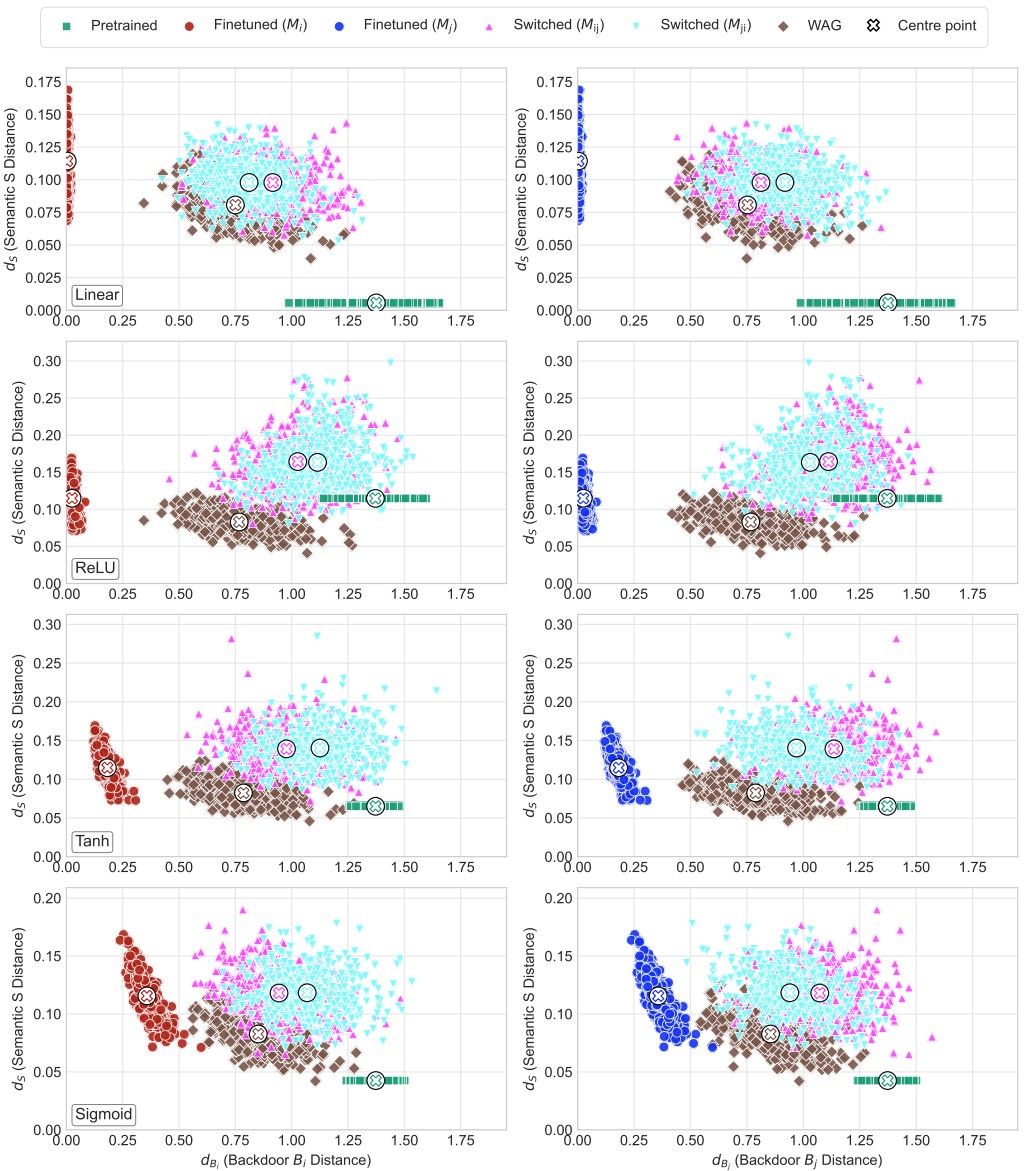

Figure 6: Euclidean distances between normalized output vectors of *pretrained*, *fine-tuned*, *WAG*, and *switched* networks, relative to semantic output $Sx$ and backdoor output $B^*x$, under *linear*, *ReLU*, *tanh*, and *sigmoid* activations.

## H  FITNESS SCORE CALCULATION FOR EVOLUTIONARY SEARCH

Building upon the heuristic rules established in Section 4.2 for disrupting backdoor connections in compromised models, we develop a comprehensive fitness function. This function incorporates five key components that collectively evaluate the quality of a module composition strategy.

### H.1  HEURISTIC RULES

Our fitness function implements the following rules through penalties and rewards:

---

**Heuristic-based Search Rules**

1. **Intra-layer adjacency penalty:** Penalizes adjacent modules from the same source model within a specific layer $i$ (*e.g.,* $Q_i$ and $K_i$).
2. **Consecutive-layer adjacency penalty:** Discourages direct connections between modules from the same source model across consecutive layers $i$ and $i+1$ (*e.g.,* $P_i$ to $Q_{i+1}$).
3. **Residual-path adjacency penalty:** Applies a distance-weighted penalty to modules from the same source model connected via residual connections between layers $i$ and $j$ (*e.g.,* $O_i$ to $Q_j$, where $j > i$), with diminishing impact as $j - i$ increases.
4. **Balance penalty:** Promotes uniform distribution of modules $\{Q, K, V, O, I, P\}$ across source models to prevent any single model from dominating the architecture.
5. **Diversity reward:** Encourages varied module combinations across layers to enhance architectural diversity.

---

### H.2  MATHEMATICAL FORMULATION

As introduced in Section 4.3, the total fitness score for a given module composition strategy $s$ is:

$$F(s) = -\lambda_1 A_{\text{intra}}(s) - \lambda_2 A_{\text{cons}}(s) - \lambda_3 A_{\text{res}}(s) - \lambda_4 B_{\text{bal}}(s) + \lambda_5 R_{\text{div}}(s), \qquad (17)$$

where all $\lambda_k$ are weight factors (default to 1.0) that control the relative importance of each component in the overall fitness score.

Each component is calculated as follows:

**1. Intra-layer Adjacency ($A_{\text{intra}}(s)$)**

$$A_{\text{intra}}(s) = \sum_{l=1}^{|s|} \text{INTRAVIOLATION}(s[l]) \qquad (18)$$

Here, INTRAVIOLATION quantifies the number of adjacent module pairs from the same source model within layer $s[l]$.

**2. Consecutive-layer Adjacency ($A_{\text{cons}}(s)$)**

$$A_{\text{cons}}(s) = \sum_{l=1}^{|s|-1} \text{CONSECVIOLATION}(s[l], s[l+1]) \qquad (19)$$

The function CONSECVIOLATION counts module pairs from the same source model that are directly connected between consecutive layers.

**3. Residual Connections ($A_{\text{res}}(s)$)**

$$A_{\text{res}}(s) = \sum_{l=1}^{|s|} \sum_{k=l+1}^{|s|} \text{RESIDUALVIOLATION}(s[l], s[k]) \times (0.5)^{k-l} \qquad (20)$$

This term evaluates residual connections between layers $s[l]$ and $s[k]$, with RESIDUALVIOLATION weighted by $(0.5)^{k-l}$ to reduce the impact of long-range connections.

**4. Module Balance ($B_{\mathbf{bal}}(s)$)**

$$B_{\text{bal}}(s) = \sum_{i=1}^{n_{\text{models}}} \sum_{m \in \mathcal{M}} |\text{count}_{i,m} - \text{count}_{\text{ideal}}| \tag{21}$$

where $\text{count}_{i,m}$ is the count of module type $m$ from model $i$, $M = \{Q, K, V, O, I, P\}$ is the set of module types, and $\text{count}_{\text{ideal}} = |s|/n_{\text{models}}$ represents the ideal count per module type per model.

**5. Layer Diversity ($R_{\mathbf{div}}(s)$)**

$$R_{\text{div}}(s) = |\text{unique}(s)| \tag{22}$$

where $\text{unique}(s)$ is the set of unique layer compositions in strategy $s$.

# I ADDITIONAL EXPERIMENT SETUP

## I.1 DATASET STATISTICS

We evaluate our method on four text and two vision datasets. The statistics of each dataset and the settings of backdoor target class are shown in Table 6. In addition, we conduct an ablation study on merging backdoored models with different target labels, and our method remains effective, as discussed in Section 5.3.

Table 6: The statistics of the evaluated text and vision datasets.

| Domain | Dataset | Classes | Train | Test Clean | Test Poison | Target Class |
|---|---|---|---|---|---|---|
| Text | SST-2 | 2 | 67,349 | 872 | 444 | Negative (0) |
| | MNLI | 3 | 100,000 | 400 | 285 | Neutral (1) |
| | AGNews | 4 | 120,000 | 7,600 | 5,700 | Sports (1) |
| Vision | CIFAR-10 | 10 | 50,000 | 10,000 | 9,000 | Automobile (1) |
| | TinyImageNet | 200 | 100,000 | 10,000 | 9,950 | European Fire Salamander (1) |

## I.2 DATASET LICENSES

We evaluate our method on the following datasets: **SST-2** (Socher et al., 2013), **MNLI** (Williams et al., 2018), **AG News** (Zhang et al., 2015), **CIFAR-10** (Krizhevsky et al., 2009), and **TinyImageNet** (Le & Yang, 2015).

The **MNLI** dataset is released under the Open American National Corpus (OANC) license, which permits free use, as stated in the original paper (Williams et al., 2018). The **AG News** dataset is distributed with a disclaimer stating it is provided "as is" without warranties and does not impose explicit restrictions on academic use.[1] No public licensing information was found for **SST-2**, **CIFAR-10**, or **TinyImageNet**. We use all datasets solely for academic, non-commercial research purposes, in accordance with standard practice in the machine learning community.

## I.3 DEFENSE BASELINES

As discussed in Section 5.1, we evaluate seven defensive approaches across text and vision domains: three model-merging techniques common to both domains, plus two domain-specific data purification methods for each–one applied during training and another during inference.

The three model-merging methods are: (1) **TIES** (Yadav et al., 2023), (2) **DARE** (Yu et al., 2024), and (3) **WAG** (Arora et al., 2024). These methods are chosen because they are applicable to both text and vision domains, do not rely on assumptions about backdoor priors, and eliminate the need for large-scale proxy clean or compromised data used for model purification or retraining. Their alignment with our setting makes them suitable for comparison. For conventional baselines, we use **Z-Def.** (He et al., 2023) and **ONION** (Qi et al., 2021a) in the text domain, which detect outlier trigger words during training and testing, respectively. For the vision domain, we select **CutMix** (Yun et al., 2019) and **ShrinkPad** (Li et al., 2021b). CutMix mitigates backdoor attacks by mixing image patches, disrupting the spatial integrity of triggers. ShrinkPad defends by shrinking the image and padding it, altering trigger placement, and reducing its effectiveness. For the vision domain, we use the BackdoorBox toolkit (Li et al., 2023b) to apply these defenses. Specifically, for CutMix, we use 30 epochs to repair the model. While these well-established methods are representative in terms of usage and performance, their dependence on data access may limit practicality in some scenarios. All baseline methods use their open-source codebases with default hyperparameters.

---

[1] http://groups.di.unipi.it/~gulli/AG_corpus_of_news_articles.html

## J    ADDITIONAL RESULTS

### J.1    OVERALL DEFENSE PERFORMANCE FOR TEXTUAL BACKDOOR ATTACKS

Due to space constraints, we present comprehensive experimental results for three datasets (**SST-2**, **MNLI**, and **AG News**) in Table 7, Table 8, and Table 9. All experiments follow the controlled settings described in Section 5.1, utilizing *RoBERTa-large* as the victim model, with results averaged across three random seeds.

We observe that our method yields decent performance on the **SST-2** dataset: it achieves top performance in 8 out of 10 attack combinations, with the remaining 2 combinations ranking second best. In cases where our method ranks first, it significantly outperforms baseline approaches. For instance, when combining BadNet with LWS attacks, our method achieves an average ASR score 21% lower than the second-best defense method. Moreover, our method consistently achieves the lowest individual ASR scores across both attacks in most combinations, highlighting its effectiveness in simultaneously mitigating multiple threats when merging compromised models.

Even in scenarios where our method ranks second, it maintains comparable defense performance to the top-performing approach. Furthermore, when combining clean models with compromised ones, our method demonstrates strong resistance against malicious attack injection, as evidenced by the lowest ASR scores. Notably, our method maintains good utility preservation across all combinations, showing minimal impact to the model performance.

Table 7: Performance comparison on the **SST-2** dataset using the *RoBERTa-large* model.

| Defense | CACC | BadNet | Insert | LWS | Hidden | AVG. | Defense | CACC | BadNet | Insert | LWS | Hidden | AVG. |
|---|---|---|---|---|---|---|---|---|---|---|---|---|---|
| Benign | 95.9 | 4.1 | 2.2 | 12.8 | 16.5 | 8.9 | Z-Def | 95.6* | 4.6 | 1.8 | 97.3 | 35.7 | 34.9 |
| Victim | 95.9* | 100.0 | 100.0 | 98.0 | 96.5 | 98.6 | ONION | 92.8* | 56.8 | 99.9 | 85.7 | 92.9 | 83.8 |
| *Combined: BadNet + InsertSent* | | | | | | | *Combined: InsertSent + LWS* | | | | | | |
| WAG | 96.3 | 56.3 | 7.4 | - | - | 31.9 | WAG | 96.1 | - | 15.1 | 43.3 | - | 29.2 |
| TIES | 95.9 | 88.7 | 17.0 | - | - | 52.9 | TIES | 96.1 | - | 35.8 | 64.9 | - | 50.3 |
| DARE | 96.5 | 57.8 | 36.3 | - | - | 47.1 | DARE | 96.4 | - | 44.4 | **31.5** | - | 37.9 |
| Ours | 96.2 | **36.9** | **7.1** | - | - | **22.0** | Ours | 96.0 | - | **11.9** | 39.7 | - | **25.8** |
| *Combined: BadNet + LWS* | | | | | | | *Combined: InsertSent + HiddenKiller* | | | | | | |
| WAG | 96.2 | 74.0 | - | 50.3 | - | 62.2 | WAG | 96.3 | - | 12.5 | - | **28.5** | 20.5 |
| TIES | 95.9 | 88.1 | - | 66.1 | - | 77.1 | TIES | 95.9 | - | 37.5 | - | 39.0 | 38.3 |
| DARE | 96.2 | 60.4 | - | 62.5 | - | 61.4 | DARE | 96.6 | - | 38.7 | - | 29.1 | 33.9 |
| Ours | 96.0 | **41.7** | - | **39.0** | - | **40.4** | Ours | 95.8 | - | **10.1** | - | 28.7 | **19.4** |
| *Combined: BadNet + HiddenKiller* | | | | | | | *Combined: LWS + HiddenKiller* | | | | | | |
| WAG | 96.1 | 63.9 | - | - | 29.0 | 46.4 | WAG | 96.4 | - | - | 60.5 | **41.7** | **51.1** |
| TIES | 96.0 | 90.4 | - | - | 36.9 | 63.6 | TIES | 96.0 | - | - | 77.8 | 55.8 | 66.8 |
| DARE | 96.7 | **36.3** | - | - | 47.6 | 41.9 | DARE | 96.7 | - | - | 67.7 | 43.3 | 55.5 |
| Ours | 96.1 | 40.5 | - | - | **27.7** | **34.1** | Ours | 96.0 | - | - | **58.6** | 47.2 | 52.9 |
| *Combined: Benign + BadNet* | | | | | | | *Combined: Benign + LWS* | | | | | | |
| WAG | 96.1 | 39.3 | - | - | - | 39.3 | WAG | 96.1 | - | - | 43.3 | - | 43.3 |
| TIES | 95.7 | 69.2 | - | - | - | 69.2 | TIES | 95.8 | - | - | 60.7 | - | 60.7 |
| DARE | 96.4 | 43.2 | - | - | - | 43.2 | DARE | 96.6 | - | - | 72.3 | - | 72.3 |
| Ours | 96.1 | **12.2** | - | - | - | **12.2** | Ours | 95.9 | - | - | **39.0** | - | **39.0** |
| *Combined: Benign + InsertSent* | | | | | | | *Combined: Benign + HiddenKiller* | | | | | | |
| WAG | 96.1 | - | 5.5 | - | - | 5.5 | WAG | 96.0 | - | - | - | **24.9** | **24.9** |
| TIES | 96.1 | - | 9.0 | - | - | 9.0 | TIES | 96.1 | - | - | - | 30.0 | 30.0 |
| DARE | 96.6 | - | 4.7 | - | - | 4.7 | DARE | 96.7 | - | - | - | 38.2 | 38.2 |
| Ours | 96.1 | - | **4.1** | - | - | **4.1** | Ours | 96.0 | - | - | - | 25.5 | 25.5 |

For the results of **MNLI** dataset Table 8, our method demonstrates more balanced and robust defense performance across different attack combinations. While DARE occasionally achieves lower ASR on individual attacks (*e.g.,* 11.6% ASR for BadNet in BadNet+InsertSent combination), it shows significant vulnerability to the other attack type (90.6% ASR for InsertSent), indicating potential risks when merging with new models. In contrast, our method maintains consistently lower average ASRs across various combinations (*e.g.,* 23.7% for BadNet+InsertSent, 43.7% for InsertSent+LWS, and 40.2% for InsertSent+Hidden), demonstrating its effectiveness in simultaneously defending against multiple attack types.

For the results of **AG NEWS** dataset Table 9, we observe a similar pattern, where our method provides more balanced defense capabilities. Notably, for the InsertSent+LWS combination, while DARE achieves a low ASR of 1.2% on LWS, it remains highly vulnerable to InsertSent attacks (99.6% ASR). In contrast, our method maintains consistently lower ASRs for both attacks (9.5% and 16.7%), resulting in a better average performance of 13.1%.

Table 8: Performance comparison on the **MNLI** dataset using the *RoBERTa-large* model.

| Defense | CACC | BadNet | Insert | LWS | Hidden | AVG. | Defense | CACC | BadNet | Insert | LWS | Hidden | AVG. |
|---|---|---|---|---|---|---|---|---|---|---|---|---|---|
| Benign | 87.6 | 12.3 | 12.6 | 26.4 | 36.9 | 22.1 | Z-Def | 89.2* | 11.1 | 11.6 | 92.2 | 50.6 | 41.4 |
| Victim | 89.5* | 100.0 | 100.0 | 96.0 | 99.9 | 99.0 | ONION | 86.3* | 64.3 | 98.6 | 89.0 | 98.8 | 87.7 |
| *Combined: BadNet + InsertSent* | | | | | | | *Combined: InsertSent + LWS* | | | | | | |
| WAG | 90.3 | 39.8 | 27.6 | - | - | 33.7 | WAG | 90.6 | - | 36.1 | 62.6 | - | 49.4 |
| TIES | 90.3 | 73.6 | 56.1 | - | - | 64.9 | TIES | 90.3 | - | 60.0 | 65.3 | - | 62.7 |
| DARE | 91.3 | 11.6 | 90.6 | - | - | 51.1 | DARE | 91.4 | - | 88.8 | 40.2 | - | 64.5 |
| Ours | 90.5 | 24.8 | 22.5 | - | - | 23.7 | Ours | 91.0 | - | 24.8 | 62.5 | - | 43.7 |
| *Combined: BadNet + LWS* | | | | | | | *Combined: InsertSent + Hidden* | | | | | | |
| WAG | 89.8 | 59.3 | - | 69.3 | - | 64.3 | WAG | 91.5 | - | 36.6 | - | 46.9 | 41.8 |
| TIES | 90.0 | 87.3 | - | 73.1 | - | 80.2 | TIES | 90.9 | - | 65.1 | - | 55.2 | 60.2 |
| DARE | 90.5 | 71.7 | - | 56.4 | - | 64.1 | DARE | 91.8 | - | 90.8 | - | 40.2 | 65.5 |
| Ours | 90.1 | 45.1 | - | 68.9 | - | 57.0 | Ours | 91.1 | - | 24.3 | - | 56.1 | 40.2 |
| *Combined: BadNet + Hidden* | | | | | | | *Combined: LWS + Hidden* | | | | | | |
| WAG | 89.9 | 61.6 | - | - | 51.7 | 56.7 | WAG | 89.8 | - | - | 70.2 | 55.1 | 62.7 |
| TIES | 90.0 | 89.4 | - | - | 64.0 | 76.7 | TIES | 90.1 | - | - | 73.8 | 59.1 | 66.5 |
| DARE | 90.9 | 33.4 | - | - | 81.8 | 57.6 | DARE | 91.0 | - | - | 41.5 | 88.7 | 65.1 |
| Ours | 90.2 | 32.5 | - | - | 59.3 | 45.9 | Ours | 89.9 | - | - | 70.3 | 57.3 | 63.8 |
| *Combined: Benign + BadNet* | | | | | | | *Combined: LWS + Benign* | | | | | | |
| WAG | 90.2 | 47.8 | - | - | - | 47.8 | WAG | 89.0 | - | - | 65.6 | - | 65.6 |
| TIES | 89.8 | 64.9 | - | - | - | 64.9 | TIES | 89.8 | - | - | 69.3 | - | 69.3 |
| DARE | 91.0 | 41.8 | - | - | - | 41.8 | DARE | 90.1 | - | - | 48.9 | - | 48.9 |
| Ours | 90.1 | 43.3 | - | - | - | 43.3 | Ours | 89.3 | - | - | 64.1 | - | 64.1 |
| *Combined: InsertSent + Benign* | | | | | | | *Combined: Hidden + Benign* | | | | | | |
| WAG | 90.4 | - | 23.2 | - | - | 23.2 | WAG | 90.3 | - | - | - | 47.0 | 47.0 |
| TIES | 90.4 | - | 40.6 | - | - | 40.6 | TIES | 89.8 | - | - | - | 54.3 | 54.3 |
| DARE | 91.3 | - | 42.3 | - | - | 42.3 | DARE | 90.9 | - | - | - | 63.3 | 63.3 |
| Ours | 90.5 | - | 18.3 | - | - | 18.3 | Ours | 89.4 | - | - | - | 47.9 | 47.9 |

## J.2 OVERALL DEFENSE PERFORMANCE FOR VISION BACKDOOR ATTACKS

Regarding the vision domain discussed in Section 5.2, we present the full results for the **CIFAR-10** and **TinyImageNet** datasets with the ViT model in Table 10 and Table 11, respectively.

While most methods achieve relatively low ASRs for many attack types, our approach is particularly effective against stealthier attacks like PhysicalBA. This is most evident in the BadNet+PhysicalBA combination, where our method reduces the ASR to 18.5% for both attacks while maintaining a high clean accuracy of 98.7% in CIFAR-10 dataset. These results highlight our method's strength in defending against more sophisticated visual backdoor attacks.

## J.3 MULTIPLE-MODEL FUSION DEFENSE

We evaluate our method in multi-model fusion scenarios as discussed in Section 5.2, beginning with three distinct backdoored models and then moving to a more challenging four-model setting involving collusion. Applied on three models, our approach derives the strategies (illustrated in Figure 15) that consistently deliver strong defensive performance, reducing the average Attack Success Rate (ASR) to below 20% across different combinations, as shown in Table 12.

Although WAG may achieve relatively low ASRs when combining multiple models, we argue that in realistic the likelihood of colluding backdoors increases. In such settings, WAG becomes less effective, as naive averaging cannot neutralize repeated malicious patterns. In contrast, our module-switching strategy is more resilient, as it strategically disrupts these recurring shortcuts. The results in Table 13, obtained using the strategy illustrated in Figure 16, confirm this advantage and demonstrate the robustness of MSD against collusive models.

Table 9: Performance comparison on the **AG NEWS** dataset using the *RoBERTa-large* model.

| Defense | CACC | BadNet | Insert | LWS | Hidden | AVG. | Defense | CACC | BadNet | Insert | LWS | Hidden | AVG. |
|---|---|---|---|---|---|---|---|---|---|---|---|---|---|
| Benign | 95.4 | 1.9 | 0.5 | 0.5 | 1.1 | 1.0 | Z-Def | 95.4* | 1.6 | 0.4 | 97.9 | 100.0 | 50.0 |
| Victim | 95.0* | 99.9 | 99.6 | 99.6 | 100.0 | 99.8 | ONION | 92.3* | 59.4 | 97.8 | 84.8 | 99.6 | 85.4 |
| *Combined: BadNet + InsertSent* | | | | | | | *Combined: InsertSent + LWS* | | | | | | |
| WAG | 95.4 | 75.2 | 60.2 | - | - | 67.7 | WAG | 95.2 | - | 39.5 | 17.8 | - | 28.7 |
| TIES | 95.3 | 92.4 | 95.6 | - | - | 94.0 | TIES | 95.1 | - | 90.5 | 55.7 | - | 73.1 |
| DARE | 95.6 | 33.7 | 66.6 | - | - | 50.1 | DARE | 95.4 | - | 99.6 | 1.2 | - | 50.4 |
| Ours | 95.3 | 72.3 | 42.5 | - | - | 57.4 | Ours | 95.1 | - | 9.5 | 16.7 | - | 13.1 |
| *Combined: BadNet + LWS* | | | | | | | *Combined: InsertSent + Hidden* | | | | | | |
| WAG | 95.2 | 76.1 | - | 28.1 | - | 52.1 | WAG | 95.4 | - | 61.4 | - | 43.6 | 52.5 |
| TIES | 95.1 | 95.6 | - | 64.4 | - | 80.0 | TIES | 95.3 | - | 93.4 | - | 75.3 | 84.4 |
| DARE | 95.4 | 99.3 | - | 3.5 | - | 51.4 | DARE | 95.5 | - | 84.0 | - | 15.8 | 49.9 |
| Ours | 95.2 | 75.8 | - | 26.0 | - | 50.9 | Ours | 95.3 | - | 41.7 | - | 47.5 | 44.6 |
| *Combined: BadNet + Hidden* | | | | | | | *Combined: LWS + Hidden* | | | | | | |
| WAG | 95.2 | 73.2 | - | - | 37.2 | 55.2 | WAG | 95.1 | - | - | 31.7 | 62.6 | 47.2 |
| TIES | 95.3 | 91.9 | - | - | 71.9 | 81.9 | TIES | 95.1 | - | - | 67.5 | 92.2 | 79.9 |
| DARE | 95.4 | 66.7 | - | - | 40.4 | 53.6 | DARE | 95.3 | - | - | 2.5 | 99.9 | 51.2 |
| Ours | 95.2 | 56.5 | - | - | 38.1 | 47.3 | Ours | 95.2 | - | - | 33.5 | 60.5 | 47.0 |
| *Combined: Benign + BadNet* | | | | | | | *Combined: Benign + LWS* | | | | | | |
| WAG | 95.4 | 65.4 | - | - | - | 65.4 | WAG | 95.2 | - | - | 14.0 | - | 14.0 |
| TIES | 95.4 | 87.4 | - | - | - | 87.4 | TIES | 95.2 | - | - | 47.1 | - | 47.1 |
| DARE | 95.6 | 33.6 | - | - | - | 33.6 | DARE | 95.6 | - | - | 2.6 | - | 2.6 |
| Ours | 95.4 | 46.4 | - | - | - | 46.4 | Ours | 95.2 | - | - | 15.7 | - | 15.7 |
| *Combined: Benign + InsertSent* | | | | | | | *Combined: Benign + Hidden* | | | | | | |
| WAG | 95.4 | - | 56.6 | - | - | 56.6 | WAG | 95.3 | - | - | - | 36.4 | 36.4 |
| TIES | 95.3 | - | 93.2 | - | - | 93.2 | TIES | 95.3 | - | - | - | 68.8 | 68.8 |
| DARE | 95.6 | - | 3.1 | - | - | 3.1 | DARE | 95.5 | - | - | - | 7.4 | 7.4 |
| Ours | 95.3 | - | 16.6 | - | - | 16.6 | Ours | 95.3 | - | - | - | 48.0 | 48.0 |

Table 10: Performance comparison on the **CIFAR-10** dataset using the *ViT* model.

| Defense | CACC | BadNet | WaNet | BATT | PBA | AVG. | Defense | CACC | BadNet | WaNet | BATT | PBA | AVG. |
|---|---|---|---|---|---|---|---|---|---|---|---|---|---|
| Benign | 98.8 | 10.1 | 10.2 | 7.7 | 10.1 | 9.5 | CutMix | 97.7* | 87.1 | 70.6 | 99.9 | 64.9 | 80.6 |
| Victim | 98.5* | 96.3 | 84.7 | 99.9 | 89.4 | 92.6 | ShrinkPad | 97.3* | 14.4 | 51.3 | 99.9 | 88.3 | 63.5 |
| *Combined: BadNet + WaNet* | | | | | | | *Combined: WaNet + BATT* | | | | | | |
| WAG | 98.7 | 13.8 | 10.6 | - | - | 12.2 | WAG | 98.7 | - | 10.2 | 22.3 | - | 16.3 |
| TIES | 98.6 | 11.9 | 10.6 | - | - | 11.3 | TIES | 98.9 | - | 10.2 | 23.9 | - | 17.0 |
| DARE | 98.8 | 83.3 | 10.2 | - | - | 46.7 | DARE | 98.9 | - | 10.2 | 45.8 | - | 28.0 |
| Ours | 98.7 | 12.3 | 10.5 | - | - | 11.4 | Ours | 98.7 | - | 10.3 | 19.1 | - | 14.7 |
| *Combined: BadNet + BATT* | | | | | | | *Combined: WaNet + PhysicalBA* | | | | | | |
| WAG | 98.9 | 10.1 | - | 42.7 | - | 26.4 | WAG | 98.8 | - | 10.2 | - | 10.2 | 10.2 |
| TIES | 98.9 | 10.1 | - | 55.8 | - | 33.0 | TIES | 98.9 | - | 10.1 | - | 10.3 | 10.2 |
| DARE | 99.0 | 69.2 | - | 26.8 | - | 48.0 | DARE | 98.9 | - | 10.1 | - | 21.0 | 15.6 |
| Ours | 98.7 | 10.2 | - | 32.6 | - | 21.4 | Ours | 98.7 | - | 10.3 | - | 10.2 | 10.2 |
| *Combined: BadNet + PhysicalBA* | | | | | | | *Combined: BATT + PhysicalBA* | | | | | | |
| WAG | 99.0 | 39.5 | - | - | 39.5 | 39.5 | WAG | 98.9 | - | - | 26.8 | 10.0 | 18.4 |
| TIES | 98.9 | 43.1 | - | - | 43.1 | 43.1 | TIES | 98.7 | - | - | 23.4 | 10.0 | 16.7 |
| DARE | 99.0 | 72.2 | - | - | 72.2 | 72.2 | DARE | 98.9 | - | - | 23.0 | 10.1 | 16.5 |
| Ours | 98.7 | 18.5 | - | - | 18.4 | 18.5 | Ours | 98.8 | - | - | 9.8 | 10.0 | 9.9 |
| *Combined: Benign + BadNet* | | | | | | | *Combined: Benign + WaNet* | | | | | | |
| WAG | 98.8 | 19.4 | - | - | - | 19.4 | WAG | 98.9 | - | 10.2 | - | - | 10.2 |
| TIES | 98.8 | 10.2 | - | - | - | 10.2 | TIES | 98.6 | - | 10.3 | - | - | 10.3 |
| DARE | 98.8 | 10.3 | - | - | - | 10.3 | DARE | 98.8 | - | 10.2 | - | - | 10.2 |
| Ours | 98.7 | 10.3 | - | - | - | 10.3 | Ours | 98.7 | - | 10.3 | - | - | 10.3 |
| *Combined: Benign + BATT* | | | | | | | *Combined: Benign + PhysicalBA* | | | | | | |
| WAG | 98.8 | - | - | 19.4 | - | 19.4 | WAG | 99.0 | - | - | - | 10.1 | 10.1 |
| TIES | 98.8 | - | - | 23.4 | - | 23.4 | TIES | 98.8 | - | - | - | 10.2 | 10.2 |
| DARE | 99.0 | - | - | 28.2 | - | 28.2 | DARE | 99.9 | - | - | - | 10.1 | 10.1 |
| Ours | 98.8 | - | - | 15.8 | - | 15.8 | Ours | 98.9 | - | - | - | 10.1 | 10.1 |

## J.4 FITNESS SCORE COMPARISON OF DIFFERENT STRATEGY

We investigate the defense performance using two different evolutionary search strategies, with and without early stopping, as illustrated in Figure 8 and 7, and present their fitness score breakdown

Table 11: Performance comparison on the **TinyImageNet** dataset using the *ViT* model.

| Defense | CACC | BadNet | WaNet | BATT | PBA | AVG. |
|---|---|---|---|---|---|---|
| Benign | 89.1 | 0.51 | 0.01 | 0.04 | 0.03 | 0.15 |
| Victim | 85.8* | 97.8 | 98.9 | 100.0 | 90.0 | 96.6 |
| *Combined: BadNet + WaNet* | | | | | | |
| WAG | 88.2 | 11.7 | 5.5 | - | - | 8.6 |
| Ours | 84.2 | **0.6** | **0.2** | - | - | **0.4** |
| *Combined: BadNet + BATT* | | | | | | |
| WAG | 87.3 | 0.11 | - | 0.15 | - | 0.13 |
| Ours | 86.8 | **0.03** | - | **0.07** | - | **0.05** |
| *Combined: BadNet + PhysicalBA* | | | | | | |
| WAG | 88.5 | 58.5 | - | - | 35.9 | 47.2 |
| Ours | 84.8 | **48.2** | - | - | **29.1** | **38.7** |

Table 12: Results of combining three backdoored models on **SST-2**. Best results are **highlighted**.

| Defense | CACC | BadNet | Insert | LWS | Hidden | AVG. | Defense | CACC | BadNet | Insert | LWS | Hidden | AVG. |
|---|---|---|---|---|---|---|---|---|---|---|---|---|---|
| *BadNet + InsertSent + LWS* | | | | | | | *BadNet + InsertSent + HiddenKiller* | | | | | | |
| WAG | 96.3 | 9.5 | **3.4** | 21.6 | - | **11.5** | WAG | 96.7 | 5.9 | 2.7 | - | 19.1 | 9.2 |
| Ours | 96.0 | **9.2** | 3.8 | 25.9 | - | 13.0 | Ours | 96.2 | 5.9 | **1.6** | - | **18.7** | **8.7** |
| *BadNet + LWS + HiddenKiller* | | | | | | | *InsertSent + LWS + HiddenKiller* | | | | | | |
| WAG | 96.0 | 10.8 | - | 30.9 | **20.3** | 20.7 | WAG | 96.0 | - | 2.7 | 25.5 | 19.6 | 15.9 |
| Ours | 96.2 | **7.9** | - | **25.7** | 20.7 | **18.1** | Ours | 96.2 | - | **2.1** | **24.1** | **19.4** | **15.2** |

in Table 14. The early stopping criterion terminates the search when no improvement in fitness score is observed over 100,000 iterations. We observe a positive correlation between the fitness score and defense performance: the adopted strategy without early stopping achieves a lower fitness score and reduces the ASR by 27.2%. By examining the score breakdowns and the visualized combinations, we attribute this improvement to fewer violations of residual connection rules in the adopted strategy, which helps disrupt subtle spurious correlations more effectively.

## J.5    STRUCTURAL OVERLAP ANALYSIS OF SEARCHED STRATEGIES

As discussed in Section 5.2, we analyze the structural overlap among the three strategies obtained from different random seeds, corresponding to the adopted strategy in Figure 7 and the two alternatives in Figure 9. The visualization is provided in Figure 10. Only 10 out of 144 module positions match across all three strategies, yielding an overlap rate of 6.94%. The overlapping positions are scattered throughout the network, and no specific region or module type exhibits higher consistency than others.

These results indicate that MSD succeeds by inducing broad structural disruption rather than depending on attack-specific or task-specific critical points, which helps make the discovered strategies transferable and reusable across different scenarios.

## J.6    RESULTS OF CANDIDATE SELECTION

As our method asymmetrically allocates modules to models, a set of candidates is generated, for which we design a selection method illustrated in Section 4.4. While the chosen candidate consistently performs well, we analyze unselected candidates' performance, as shown in Table 15. Our selection method correctly identifies the best candidates in most cases, outperforming alternatives by a significant margin. Although some unselected candidates achieve a lower ASR in certain cases, our selected candidate maintains comparable performance.

Table 13: Performance comparison in a four-model fusion scenario with backdoor *collusion* on the **SST-2** dataset using the **roberta-large** model. We combine two pairs of models, where each pair shares the same backdoor attack. Best defensive results (lowest ASR) are **highlighted**.

| Combination | Defense | CACC (%) (↑) | ASR (↓) Atk1 | Atk2 | AVG. |
|---|---|---|---|---|---|
| BadNet+BadNet+Sent+Sent | WAG | 96.3 | 56.3 | **7.4** | 31.9 |
| | Ours (MSD) | 96.7 | **32.0** | 9.1 | **20.5** |
| BadNet+BadNet+LWS+LWS | WAG | 96.1 | 74.0 | **50.3** | 62.2 |
| | Ours (MSD) | 95.7 | **51.1** | 60.3 | **55.7** |
| BadNet+BadNet+Hidden+Hidden | WAG | 96.2 | 63.9 | 29.0 | 46.4 |
| | Ours (MSD) | 96.2 | **42.4** | **24.2** | **33.3** |

Table 14: Comparison of strategy fitness scores and performance in combining *Benign* with *BadNet* model.

| Early Stopping Strategy | | Adopted Strategy | |
|---|---|---|---|
| *Fitness Score Components* | | | |
| Intra Layer Score | **-42.00** | Intra Layer Score | -48.00 |
| Inter Layer Score | -21.00 | Inter Layer Score | **-15.00** |
| Residual Connection Score | -48.24 | Residual Connection Score | **-24.02** |
| Balance Score | 0.00 | Balance Score | 0.00 |
| Diversity Score | **17.00** | Diversity Score | 12.00 |
| **Total Score** | **-94.24** | **Total Score** | **-75.01** |
| *Performance Metrics* | | | |
| CACC (↑) | 96.70 | CACC (↑) | 96.10 |
| ASR (↓) | 39.40 | ASR (↓) | **12.20** |

## J.7 IMPORTANCE OF HEURISTIC RULES

We introduce five heuristic rules in Section 4.2 to guide the evolutionary search for module switching strategies. To assess the contribution of each rule, we perform ablation experiments by individually removing the first three rules, which aim to disconnect adjacent modules at different structural levels, and measure the resulting defense performance under three settings. As shown in Table 16, removing any of these rules generally leads to performance degradation, supporting the complementary nature of the full rule set. We further visualize the searched strategies resulting from each ablation in Figures 11 to 13.

## J.8 GENERALIZATION ACROSS MODEL ARCHITECTURES

As discussed in Section 5.3, we evaluate our method across three model architectures–*RoBERTa-large*, *BERT-large*, and *DeBERTa-v3-large*–under three backdoor settings. As shown in Table 17, our defense consistently achieves lower ASR compared to the baseline WAG across all models. Notably, we apply the same unified searched strategy (presented in Figure 7) to all architectures, demonstrating the strong generalization and transferability of our method. This supports its scalability and practicality in real-world applications.

To further examine the generality of MSD beyond Transformer-based families, we extend MSD to CNN architectures, ResNet-18 and ResNet-50 (He et al., 2016), on the CIFAR-10 dataset. The corresponding searched strategies are presented in Figure 17 and Figure 18, and the quantitative results are shown in Table 18.

Across all attack combinations, MSD provides robust defense performance that is comparable to or better than WAG. For ResNet-18, MSD achieves average ASRs of 11.78% and 10.59% under the

Table 15: Performance comparison of selected and unselected candidates on **SST-2**.

| Setting | Selection candidate | | Unselected candidate | | Overall Mean ASR ($\downarrow$) | WAG Mean ASR ($\downarrow$) |
|---------|---------------------|---|----------------------|---|---------------------------------|-----------------------------|
| | CACC ($\uparrow$) | AVG. ASR ($\downarrow$) | CACC ($\uparrow$) | AVG. ASR ($\downarrow$) | | |
| BadNet+InsertSent | 96.2 | **22.0** | 96.5 | 31.2 | 26.6 | 31.9 |
| BadNet+LWS | 96.0 | **40.4** | 95.9 | 72.4 | 56.4 | 62.2 |
| BadNet+Hidden | 96.1 | **34.1** | 96.0 | 48.5 | 41.3 | 46.5 |
| InsertSent+LWS | 96.0 | **25.8** | 96.0 | 30.3 | 28.1 | 29.2 |
| InsertSent+Hidden | 95.8 | 19.4 | 96.1 | **19.2** | 19.3 | 20.5 |
| LWS+Hidden | 96.0 | 52.9 | 96.2 | **49.6** | 51.3 | 51.1 |
| Average | 96.0 | **32.4** | 96.1 | 41.9 | 37.2 | 40.2 |

Table 16: Impact of heuristic rule ablations under different combinations of backdoor settings on **SST-2** using the *RoBERTa-large* model. $\Delta$ denotes the change in average ASR relative to the full rule set.

| Setting | Ablation | CACC ($\uparrow$) | ASR ($\downarrow$) | | | |
|---------|----------|-------------------|------|------|------|---|
| | | | Atk1 | Atk2 | AVG. | $\Delta$ |
| BadNet + InsertSent | All rules (full) | 96.2 | 36.9 | 7.1 | **22.0** | – |
| | w/o rule 1 | 96.0 | **33.2** | 18.7 | 25.9 | +3.9 |
| | w/o rule 2 | 96.3 | 60.6 | 14.1 | 37.3 | +15.3 |
| | w/o rule 3 | 96.3 | 43.1 | **6.2** | 24.6 | +2.6 |
| BadNet + LWS | All rules (full) | 96.0 | **41.7** | **39.0** | **40.4** | – |
| | w/o rule 1 | 95.9 | 46.2 | 51.2 | 48.7 | +8.3 |
| | w/o rule 2 | 96.0 | 68.1 | 62.8 | 65.4 | +25.0 |
| | w/o rule 3 | 96.0 | 69.1 | 46.3 | 57.7 | +17.3 |
| BadNet + Hidden | All rules (full) | 96.1 | 40.5 | **27.7** | 34.1 | – |
| | w/o rule 1 | 95.9 | **14.0** | 32.8 | **23.4** | -10.7 |
| | w/o rule 2 | 96.1 | 59.4 | 29.4 | 44.4 | +10.3 |
| | w/o rule 3 | 96.0 | 56.6 | 29.1 | 42.9 | +8.8 |

BadNet+BATT and BadNet+WaNet settings, outperforming WAG while keeping clean accuracy stable. For ResNet-50, MSD reduces the average ASR to 11.07% and 9.90% on the two combinations, again achieving the lowest ASRs among all evaluated methods.

These results indicate that MSD naturally extends to convolutional architectures and maintains strong defensive capability without requiring CNN-specific modifications, supporting its cross-domain applicability.

### J.9 MINIMUM CLEAN DATA REQUIREMENT

By default, we use 50 clean data points per class to guide the candidate selection process (as described in Section 4.4). To further investigate the minimum clean data required for effective defense, we reduce this to 20 samples per class across all three model architectures on SST-2. As shown in Table 17, our approach continues to select candidates with low ASR even under this constrained setting. These results indicate that the method remains effective in low-resource scenarios with limited clean supervision.

### J.10 PERFORMANCE UNDER VARYING POISONING RATES

As discussed in Section 5.3, we further evaluate the robustness of our method under varying poisoning rates (20%, 10%, and 1%) on SST-2 dataset using the *RoBERTa-large* model. As shown

Table 17: Cross-model evaluation under varying clean data budgets on **SST-2**. $N = 50$ and $N = 20$ indicate the number of clean samples per class used for validation.

| Setting | Defense | RoBERTa-large | | | | BERT-large | | | | DeBERTa-v3-large | | | |
|---|---|---|---|---|---|---|---|---|---|---|---|---|---|
| | | CACC ($\uparrow$) | ASR ($\downarrow$) | | | CACC ($\uparrow$) | ASR ($\downarrow$) | | | CACC ($\uparrow$) | ASR ($\downarrow$) | | |
| | | | Atk1 | Atk2 | AVG. | | Atk1 | Atk2 | AVG. | | Atk1 | Atk2 | AVG. |
| BadNet + InsertSent | WAG | 96.3 | 56.3 | 7.4 | 31.9 | 93.3 | 40.2 | 60.1 | 50.2 | 96.1 | 47.4 | 5.2 | 26.3 |
| | Ours ($N = 50$) | 96.2 | 36.9 | 7.1 | 22.0 | 93.5 | 39.7 | 38.1 | 38.9 | 96.3 | 40.4 | 5.2 | 22.8 |
| | Ours ($N = 20$) | 96.2 | 47.7 | 6.6 | 27.1 | 93.5 | 39.7 | 38.1 | 38.9 | 96.3 | 32.8 | 5.1 | 19.0 |
| BadNet + LWS | WAG | 96.2 | 74.0 | 50.3 | 62.2 | 93.1 | 76.9 | 63.0 | 69.9 | 96.2 | 63.4 | 79.5 | 71.5 |
| | Ours ($N = 50$) | 96.0 | 41.7 | 39.0 | 40.4 | 93.0 | 73.9 | 61.3 | 67.6 | 96.0 | 48.7 | 73.0 | 60.8 |
| | Ours ($N = 20$) | 96.0 | 41.7 | 39.0 | 40.4 | 93.0 | 76.5 | 63.6 | 70.0 | 96.0 | 48.7 | 73.0 | 60.8 |
| BadNet + Hidden | WAG | 96.1 | 63.9 | 29.0 | 46.5 | 93.3 | 56.9 | 43.8 | 50.3 | 96.2 | 48.3 | 39.6 | 43.9 |
| | Ours ($N = 50$) | 96.1 | 40.5 | 27.7 | 34.1 | 93.4 | 50.3 | 37.9 | 44.1 | 96.1 | 22.7 | 41.0 | 31.8 |
| | Ours ($N = 20$) | 96.2 | 34.9 | 25.6 | 30.3 | 93.4 | 50.3 | 37.9 | 44.1 | 96.3 | 22.7 | 41.0 | 31.8 |

Table 18: Defense performance on CNN architectures (**ResNet-18** and **ResNet-50**) on CIFAR-10. MSD achieves comparable or lower ASRs than WAG across diverse combinations.

| Model | Combination | Method | CACC ($\uparrow$) | Atk1 ASR ($\downarrow$) | Atk2 ASR ($\downarrow$) | AVG. ASR ($\downarrow$) |
|---|---|---|---|---|---|---|
| ResNet-18 | BadNet + BATT | No Defense | 95.93* | 98.34 | 99.84 | 99.09 |
| | | WAG | 96.34 | 10.21 | 13.87 | 12.04 |
| | | Ours | 94.46 | 10.18 | 13.37 | 11.78 |
| | BadNet + WaNet | No Defense | 95.79* | 98.34 | 100.0 | 99.17 |
| | | WAG | 96.14 | 10.94 | 10.15 | 10.55 |
| | | Ours | 94.41 | 11.26 | 9.91 | 10.59 |
| ResNet-50 | BadNet + BATT | No Defense | 95.13* | 98.42 | 99.81 | 99.12 |
| | | WAG | 96.61 | 10.29 | 13.71 | 12.00 |
| | | Ours | 95.59 | 10.02 | 12.11 | 11.07 |
| | BadNet + WaNet | No Defense | 94.99* | 98.42 | 99.91 | 99.17 |
| | | WAG | 96.53 | 10.31 | 9.99 | 10.15 |
| | | Ours | 96.13 | 10.04 | 9.75 | 9.90 |

in Table 19, our method consistently achieves lower ASR than WAG across settings that combine models poisoned with different attack methods and poisoning ratios.

## J.11 PERFORMANCE UNDER ADAPTIVE ATTACKS

As discussed in Section 5.3, we evaluate robustness under two adaptive scenarios: (1) when a searched strategy is exposed and exploited by an attacker who adversarially retrains the model by freezing unused modules while continuing to fine-tune the selected ones on poisoned data, aiming to preserve them after module switching; (2) advanced adaptive backdoor attacks such as Adaptive-Patch (Qi et al., 2023).

For (1), there is no single fixed strategy: the defender can rerun the search with different seeds to obtain diverse strategies. This flexibility provides protection even if the attacker has adapted to a known one. For example, when the adversary adapts to Strategy Figure 7, alternative strategies from different seeds (Figure 9) still mitigate the attack. We simulate an attacker aware of Strategy 1 (S1) and defend using Strategy 2 (S2) and Strategy 3 (S3), with results on SST-2 (*RoBERTa-large*) reported in Table 20.

For (2), we evaluate the **Adaptive-Patch** attack (Qi et al., 2023). Following the transferability strategy in Figure 14, our method consistently demonstrates strong performance against this more challenging setting, as shown in Table 21.

## J.12 PERFORMANCE UNDER LABEL-INCONSISTENT AND IDENTICAL BACKDOOR ATTACKS

A practical consideration for model merging is that the obtained models may be trained by different attackers targeting different labels, or they may encode identical backdoor attacks. We therefore examine two challenging settings: (1) models with inconsistent target labels, and (2) models trained with the same backdoor. In the first case, where each model has a different target label, our method

Table 19: Performance comparison under varying poison rates on **SST-2** using the *RoBERTa-large* model.

| Setting | Defense | Poison Rate: 20% | | | | Poison Rate: 10% | | | | Poison Rate: 1% | | | |
|---|---|---|---|---|---|---|---|---|---|---|---|---|---|
| | | CACC (↑) | ASR (↓) | | | CACC (↑) | ASR (↓) | | | CACC (↑) | ASR (↓) | | |
| | | | Atk1 | Atk2 | AVG. | | Atk1 | Atk2 | AVG. | | Atk1 | Atk2 | AVG. |
| BadNet + InsertSent | WAG | 96.3 | 56.3 | 7.4 | 31.9 | 96.1 | 66.6 | **8.9** | 37.9 | 96.4 | 58.3 | **27.2** | **42.8** |
| | Ours (MSD) | 96.2 | **36.9** | **7.1** | **22.0** | 96.0 | **55.1** | 9.3 | **32.3** | 96.3 | **57.4** | 44.4 | 50.9 |
| BadNet + LWS | WAG | 96.2 | 74.0 | 50.3 | 62.2 | 95.1 | 83.7 | 46.3 | 65.0 | 96.3 | 62.7 | 28.9 | 45.8 |
| | Ours (MSD) | 96.0 | **41.7** | **39.0** | **40.4** | 94.9 | **70.6** | **40.1** | **55.3** | 96.4 | **59.9** | **27.6** | **43.7** |
| BadNet + Hidden | WAG | 96.1 | 63.9 | 29.0 | 46.5 | 95.9 | 67.9 | 26.9 | 47.4 | 96.1 | 64.9 | 30.5 | 47.7 |
| | Ours (MSD) | 96.1 | **40.5** | **27.7** | **34.1** | 95.5 | **51.9** | **25.8** | **38.9** | 96.1 | **59.2** | **30.0** | **44.6** |

Table 20: Robustness against attacker adapted to Strategy 1 (S1) on SST-2 with *RoBERTa-large*.

| No. | Defense | Variant | CACC (↑) | ASR (↓) | | |
|---|---|---|---|---|---|---|
| | | | | Atk1 | Atk2 | Avg |
| 1 | No Defense | BadNet | 96.0 | 100.0 | – | – |
| 2 | No Defense | InsertSent | 96.3 | – | 100.0 | – |
| 3 | No Defense | BadNet (adaptive, S1 known) | 95.6 | 100.0 | – | – |
| 4 | | WAG | 96.7 | 56.3 | 7.4 | 31.9 |
| 5 | Merge 1+2 | Ours (w/ S2) | 96.0 | **39.0** | **8.0** | **23.5** |
| 6 | | Ours (w/ S3) | 96.0 | **39.4** | 20.6 | **30.0** |
| 7 | | WAG | 95.8 | 59.8 | 9.2 | 34.5 |
| 8 | Merge 2+3 | Ours (w/ S2) | 96.1 | **21.3** | **7.2** | **14.3** |
| 9 | | Ours (w/ S3) | 96.0 | **57.5** | **5.4** | **31.5** |

maintains strong defensive performance. In the second case, when models are attacked with the same method but different labels, our method again substantially reduces ASR compared to WAG, as shown in Table 22.

Table 21: Defense performance under Adaptive-Patch attacks on CIFAR-10 using *ViT*.

| Setting | Defense | CACC (↑) | ASR (↓) | | |
|---|---|---|---|---|---|
| | | | Atk1 | Atk2 | Avg |
| BadNet | | 98.4 | 97.3 | – | – |
| BATT | No Defense | 98.4 | 99.4 | – | – |
| Adaptive Patch | | 98.0 | – | 86.1 | – |
| BadNet + Adaptive Patch | WAG | 98.5 | 24.3 | 17.6 | 20.9 |
| | Ours | 98.5 | **13.9** | **16.5** | **15.2** |
| BATT + Adaptive Patch | WAG | 98.7 | **0.8** | 10.3 | **5.5** |
| | Ours | 98.6 | 1.3 | 10.3 | 5.8 |

Table 22: Results under inconsistent-target-label and identical backdoor attacks cases.

| Dataset | Combination | Method | CACC (↑) | ASR (↓) | | |
|---|---|---|---|---|---|---|
| | | | | Atk1 | Atk2 | Avg |
| *Label-Inconsistent Cases* | | | | | | |
| SST-2 | BadNet (label=0) + InsertSent (label=1) | WAG | 96.4 | 47.2 | 71.2 | 59.2 |
| | | Ours | 96.2 | **20.0** | **65.9** | **43.0** |
| CIFAR-10 | BadNet (label=1) + BATT (label=2) | WAG | 98.7 | **0.3** | **18.8** | **9.6** |
| | | Ours | 98.6 | 0.4 | 19.1 | 9.8 |
| *Identical-Attack Cases (Different Labels)* | | | | | | |
| SST-2 | BadNet (label=0) + BadNet (label=1) | WAG | 96.3 | 78.0 | **2.4** | 40.2 |
| | | Ours | 96.1 | **38.7** | 16.0 | **27.4** |

## K   EXAMPLES OF SEARCHED STRATEGIES

We present several representative examples of module-switching strategies discovered by our evolutionary algorithm, grouped below for clarity.

- **Two-model strategies on Transformer and ViT architectures.**
    - Adopted merging strategy for two *RoBERTa-large* models (24 layers), shown in Figure 7, with a fitness score of -75.0.
    - An early-stage merging strategy for two *RoBERTa-large* models, illustrated in Figure 8, yielding a fitness score of -94.2.
    - Two alternative full-search strategies obtained with distinct random seeds, shown in Figure 9, achieving fitness scores of -76.5 and -72.0.
    - Structural overlap analysis across these three strategies (Figure 10), showing that only 6.94% of module positions coincide, indicating high structural diversity.
    - Adopted strategy for merging two 12-layer models (*e.g., ViT*), presented in Figure 14, with a fitness score of -39.5.

- **Strategies under ablation and multi-model fusion settings.**
    - Strategies derived by ablating individual heuristic rules (Figures 11–13), used to assess the contribution of each rule.
    - Adopted strategy for merging three *RoBERTa-large* models (24 layers), depicted in Figure 15, with a fitness score of -26.2.
    - Adopted strategy for merging four models, shown in Figure 16, with a fitness score of -11.0.

- **Strategies on CNN architectures.**
    - Searched merging strategy for two *ResNet-18* models, shown in Figure 17, with a fitness score of -1.4.
    - Searched merging strategy for two *ResNet-50* models, shown in Figure 18, with a fitness score of -1.9.

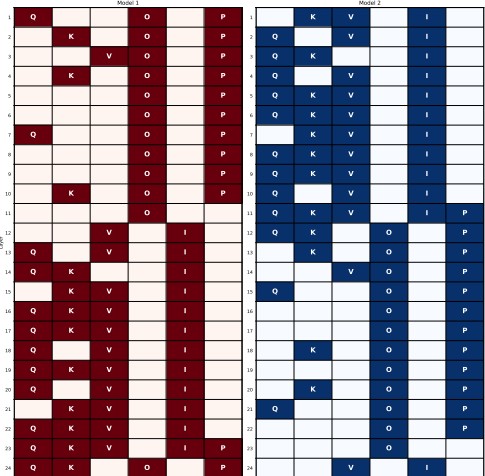

Figure 7: Adopted merging strategy (with a fitness score of -75.0).

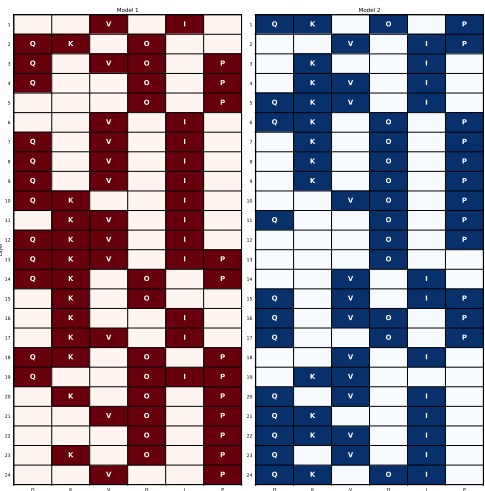

Figure 8: Early stopping strategy (with a fitness score of -94.2).

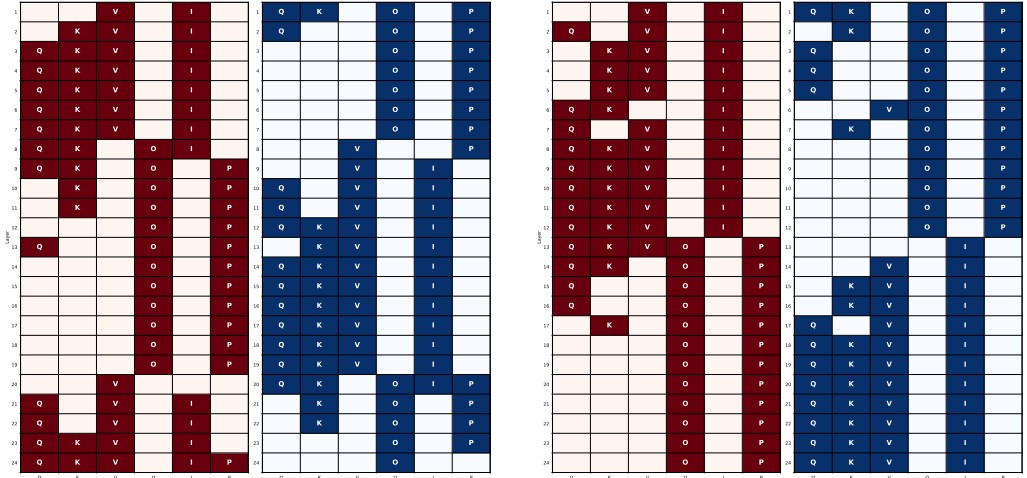

Figure 9: Two alternative discovered full search merging strategies using distinct random seeds. The strategies yielded comparable fitness scores of -76.5 and -72.0, respectively.

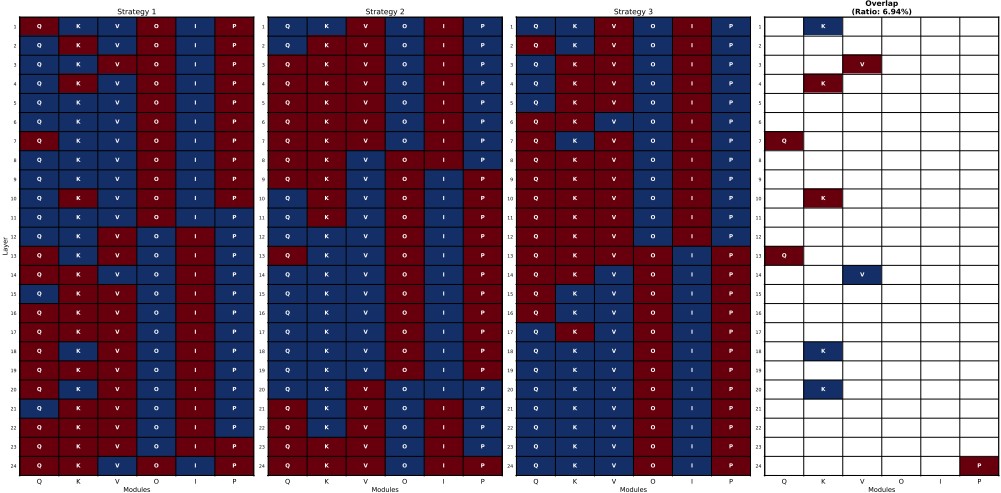

Figure 10: Overlap analysis across three strategies (Figure 7 and Figure 9). Only 6.94% of module positions coincide across all strategies, indicating high structural diversity.

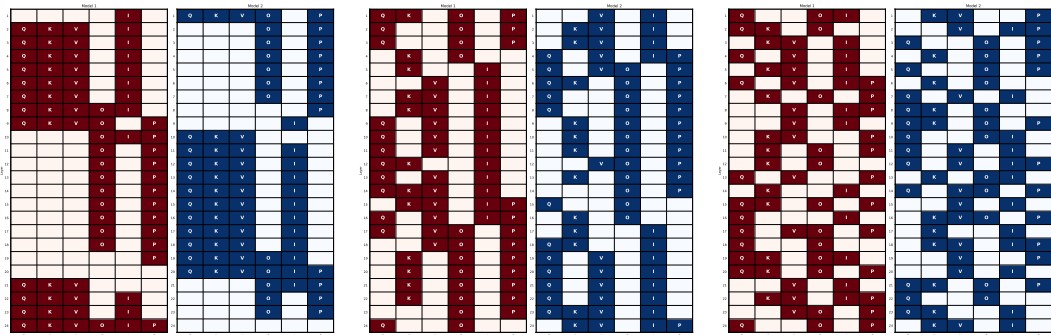

Figure 11: Strategy of Ablating rule 1.

Figure 12: Strategy of ablating rule 2.

Figure 13: Strategy of ablating rule 3.

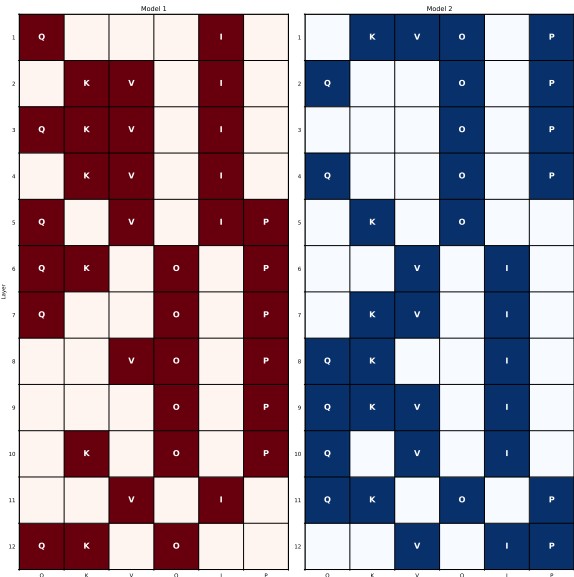

Figure 14: Adopted strategy for merging two 12-layer models (*e.g., ViT*) (fitness score: -39.5).

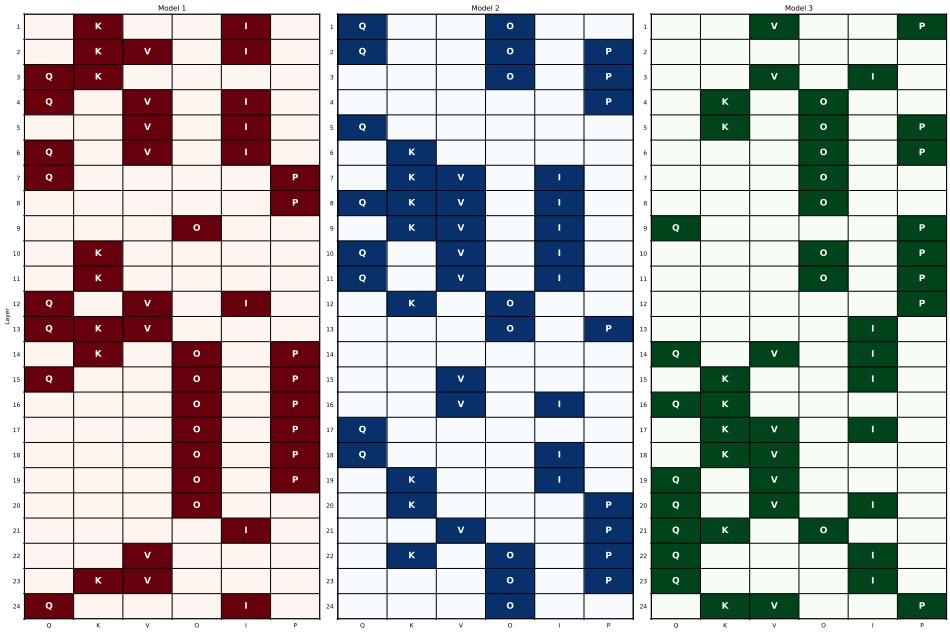

Figure 15: Adopted merging strategy for merging three models (fitness score -26.2).

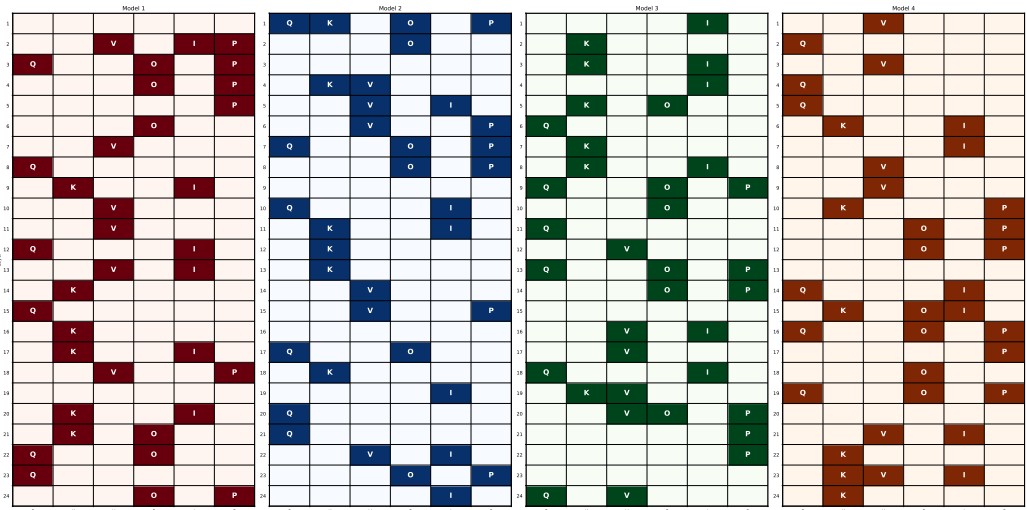

Figure 16: Adopted strategy for merging four models (fitness score -11.0).

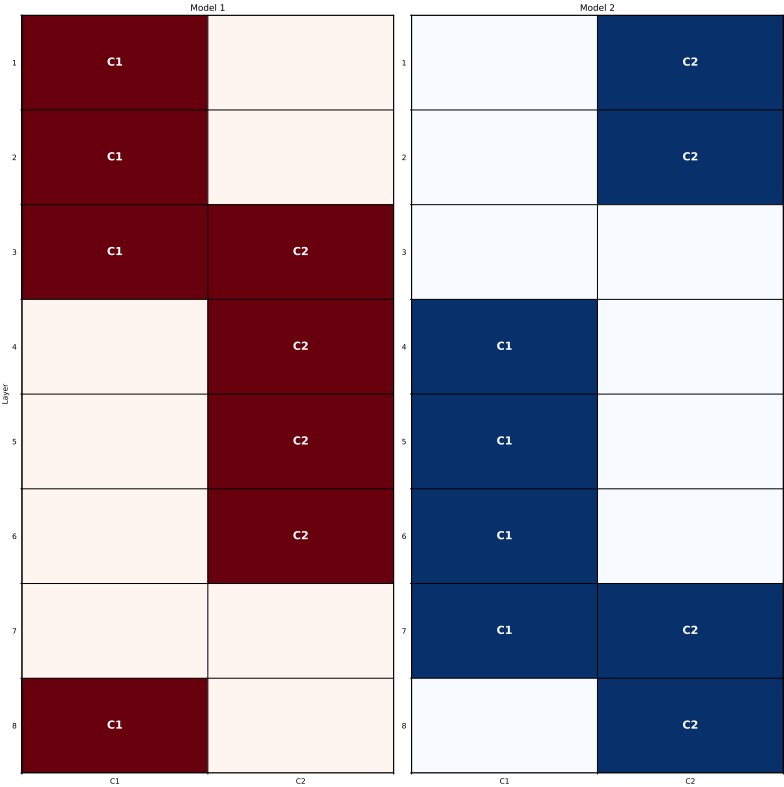

Figure 17: Searched merging strategy for two *ResNet-18* models (fitness score: -1.4).

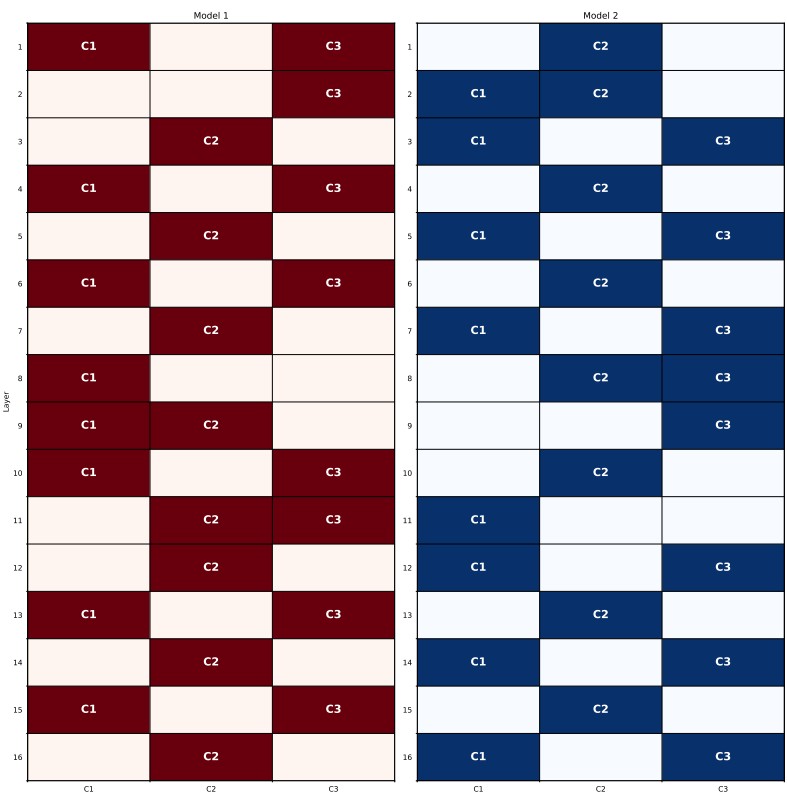

Figure 18: Searched merging strategy for two *ResNet-50* models (fitness score: -1.9).

