# OpenReview forum: "Defending against Backdoor Attacks via Module Switching"
_ICLR.cc/2026/Conference — ICLR 2026 Poster_

### Official Review · Reviewer_W5FL · 2025-10-21

**Soundness:** 2
**Presentation:** 2
**Contribution:** 2
**Rating:** 2
**Confidence:** 5

**Summary:**

The paper proposes a "model merging" approach to mitigating backdoors in neural networks. The paper works with different neural networkw with identical or very similar model architectures trained on the same type of data.
Experiments are conducted on NLP (BERT) and ViT models.

**Strengths:**

Addressing backdoors/Trojans is a challenging problem.

**Weaknesses:**

There is a large body of work on backdoor defense, but
the literature review at about line 054 is terse and odd.
Many of the most potent backdoor defenses are post-training leveraging a small benign dataset. Papers in this setting like Neural Cleanse and I-BAU are not discussed (though this assumption is made on line 240). It's not clear what is meant by "although compromised auxiliary models can be used as defensive references". The original IARPA TrojAI solicitation on backdoor defense suggested the defender could use known-clean models and known-attacked models. But this scenario begs the question: How are these models identified without assuming the action of some unspecified backdoor detector or poisoning models with _known_ backdoor attacks? That is, this is a supervised defensive scenario.

Re. Section 3, it's not clear to me why two-layer neural networks with additional assumptions are interesting.

At a high level, if all the models are likewise poisoned, I don't see how the proposed approach will be effective (noting that an effective backdoor attack will not affect the performance on benign examples available to the defender).
The advocated evolutionary search approach can be quite computationally complex.

Do they discuss the impact on accuracy of this method?

Re. backdoored models prioritize trigger features (line 313): The authors should also cite [H. Wang et al.  MM-BD...] there. It's not clear why defense baselines don't also include such methods (MM-BD, I-BAU, etc.) which I think would be a lot less computationally complex.

Though the authors do discuss the amount of clean data required (line 458 with details in Appendix), I think the amount of data used for different methods could be reported in their performance results.  Moreover, in the body  of the paper the authors should report the computational costs of their method and those compared against.  I would have reported both the amount of data and computational costs should be in the body of the paper while the "theory" of Sec. 3 should be in the Appendix.

**Questions:**

See above.

---

> ### Author Response · Authors · 2025-11-21
> **Response to Reviewer W5FL (1/3)**
>
> We thank the reviewer for the time and effort spent on the review. We wish to briefly note that some concerns appear to rely on interpretations that differ from the intended problem setting, the role of our theoretical analysis, and the scope of the related work in our paper. We address all points raised below.
>
> ---
>
> > **W1:** It is not clear why two-layer neural networks with additional assumptions are interesting, and the theoretical analysis of Section 3 should be in the appendix.
>
> **A1:** We respectfully disagree with the premise that theoretical analysis on simplified networks is uninteresting or should be moved to the appendix. Analyzing two-layer networks is a well-established methodology in the deep learning community for isolating key mechanisms that are difficult to study directly in full-scale architectures.
>
> We believe retaining Section 3 in the main paper is important for evaluating the scientific contribution of our work, for two reasons:
> 1. **Alignment with established community practice.**
>    The use of two-layer or otherwise simplified architectures for theoretical insight is generally useful in machine learning work. Representative examples include studies on optimization, adversarial robustness, and backdoor learning:
>   [A] Anthropic (2021): A Mathematical Framework for Transformer Circuits
>   [B] ICML 2024: A Theoretical Analysis of Backdoor Poisoning Attacks in CNNs
>   [C] NeurIPS 2023: Adversarial Examples Exist in Two-Layer ReLU Networks for Low Dimensional Linear Subspaces
>   [D] NeurIPS 2025: Alternating Gradient Flows: A Theory of Feature Learning in Two-layer Neural Networks
>   [E] ICLR 2025: On the Optimization and Generalization of Two-layer Transformers with Sign Gradient Descent
>     These works demonstrate that simplified analyses remain a valuable and widely used tool for understanding modern deep models.
>
> 2. **Theoretical foundation of our method.**
>    Section 3 is not a detached component; it provides the formal reasoning explaining why module switching disrupts backdoor propagation while preserving pretrained semantics, using an output-distance-based characterization. This analysis directly motivates the algorithmic design in Section 4. Moving it to the appendix would weaken the conceptual continuity between our theoretical motivation and the resulting method.
>
> For these reasons, we believe Section 3 belongs in the main paper as part of the core contribution.
>
> ---
>
> > **W2:** The literature review around line 054 is terse and odd; many of the most potent post-training backdoor defenses that leverage a small benign dataset (such as Neural Cleanse and I-BAU) are not discussed.
>
> **A2:** We wish to clarify that the paragraph at line 054 is not intended as a survey of all post-training backdoor defenses. Its purpose is to highlight the constraints of model-merging-based defenses specifically. Neural Cleanse and I-BAU are not part of this category, which is why they are not discussed in that context.
>
> However, we have included them in our related work section: Neural Cleanse (line 117, Wang et al., 2019) and I-BAU (line 121, Zeng et al., 2022). We are glad to clarify the distinction in the related work section to avoid any potential ambiguity.
>
> ---
>
> > **W3:** It’s not clear what is meant by “although compromised auxiliary models can be used as defensive references” in line 056; scenarios like the IARPA TrojAI solicitation assume access to known-clean models and known-attacked models, which raises the question of how such models would be identified without assuming an unspecified backdoor detector or poisoning models with known backdoor attacks, resulting in a supervised defensive scenario.
>
> **A3:** We clarify that the phrase in line 056 refers to prior auxiliary-model-based defenses (e.g., Li et al., 2024; Tong et al., 2024) that use a “reference model” to guide purification. As noted in our text, using such references-including potentially compromised ones-can introduce additional risks (He et al., 2025, table 5), and this constraint directly motivates our setting.
>
> In contrast, MSD assumes that all available models may be untrusted and may contain different backdoors. We do not assume access to known-clean or known-attacked models, nor do we rely on backdoor detectors or supervised labeling. For this reason, our setting is fundamentally different from the IARPA TrojAI scenario mentioned by the reviewer.

---

> > ### Author Response · Authors · 2025-11-21
> > **Response to Reviewer W5FL (2/3)**
> >
> > > **W4:** If all the models are likewise poisoned, it is not clear how the proposed approach will be effective, noting that an effective backdoor attack will not affect the performance on benign examples available to the defender.
> >
> > **A4:** We identify two possible interpretations of the reviewer’s concern regarding “likewise poisoned” models:
> > (a) If backdoors do not degrade benign accuracy, how can MSD avoid selecting poisoned modules when only a benign validation set is available?
> > (b) Whether the method remains effective if all source models share completely identical backdoors.
> >
> > We hope these two aspects capture the intended scope. If we have misinterpreted the setting, we would welcome further clarification.
> >
> > **Regarding (a):** MSD does not attempt to detect which modules are poisoned. Unlike pruning or unlearning methods, MSD relies on the intuition that backdoors operate as learned shortcut pathways, and that disrupting these shortcut dependencies through module switching weakens backdoor propagation regardless of where the backdoor resides.
> > This is exactly what Section 3 formalizes: our theoretical justification shows that module switching induces larger deviations in backdoor-related signals than weight averaging, explaining why MSD remains effective even when all models are poisoned.
> >
> > **Regarding (b):** We analyze this “collusive” scenario in Section 5.2 (Line 429). As shown in Table 12, even when models share identical backdoor knowledge, MSD still provides meaningful defense, while WAG degrades noticeably. We acknowledge that in the extreme case where all models share the exact same backdoor pattern and location, the backdoor effectively becomes shared knowledge and is inherently difficult to mitigate. We view addressing this fully collusive setting as a promising direction for future work.
> >
> > ---
> >
> > > **W5:** The advocated evolutionary search approach can be quite computationally complex, and the authors should report the computational costs of their method and those compared against.
> >
> > **A5:** We have quantified the computational cost of MSD and compared it with the key baselines (DARE, TIES, WAG) in the two-model merging setting. The results are:
> > | Phase | DARE | TIES | WAG | MSD |
> > | --- | --- | --- | --- | --- |
> > | Search | 2.5 hrs | - | - | 2.6 hrs |
> > | Merge | - | 1 min | 10 s | 16 s |
> >
> > Note that the search cost is **architecture-dependent only** and can be performed **once offline**. After a strategy is obtained, it can be reused for any models with the same structure, reducing the **deployment-time cost to 16 seconds**, which is comparable to WAG.
> >
> > This is in contrast to deployment-time search methods such as DARE, which require roughly 2.5 hours for every merge.
> >
> > We will include this cost analysis in the revised manuscript to provide a clear and transparent account of the computational complexity.
> >
> > ---
> >
> > > **W6:** Does the paper discuss the impact of the proposed method on accuracy?
> >
> > **A6:** Yes. We evaluate the impact on accuracy both theoretically and empirically. In Section 3, we discuss the utility loss of module switching and show that the operation does not increase the overall deviation from benign semantics. Our two-layer experiments (line 239) further confirm that the per-model utility loss remains small. For deep models, **ALL** experimental tables (Tables 1-2 and 6-20) report clean accuracy alongside ASR, and MSD consistently preserves accuracy at a level comparable to the baselines while providing stronger backdoor mitigation.

---

> > > ### Author Response · Authors · 2025-11-21
> > > **Response to Reviewer W5FL (3/3)**
> > >
> > > > **W7:** For the statement “backdoored models prioritize trigger features” (line 313), the author should also cite MM-BD (H. Wang et al.), and it is not clear why baselines do not include such methods (MM-BD, I-BAU, etc.), which may be less computationally complex.
> > >
> > > **A7:** We thank the reviewer for the suggestion and will add the citation to MM-BD in line 313. Note that MM-BD (line 119) and I-BAU (line 121) are already included in our related work.
> > >
> > > We excluded these methods as baselines for two reasons:
> > > * **Orthogonal Problem Settings:** MM-BD and I-BAU focus on single-model defenses (either through detection or unlearning). In contrast, our work studies how to combine multiple available models to disrupt backdoors. Accordingly, we compare against baselines designed for the multi-model setting, such as WAG, TIES, and DARE. We view MM-BD and I-BAU as orthogonal and complementary to our merging-based approach rather than direct competitors.
> > > * **Architecture Mismatch:** As noted in line 333, our primary evaluation is in the NLP domain to align with the WAG baseline on Transformer models. For cross-domain vision experiments, we naturally adopt ViT models. MM-BD and I-BAU were developed and evaluated mainly on CNN architectures, and adapting them to ViTs would require substantial effort and is outside the scope of this work.
> > >
> > > These methods are therefore methodologically orthogonal rather than competing baselines. We will clarify this distinction in the revision.
> > >
> > > ---
> > >
> > > > **W8:** Though the authors do discuss the amount of clean data required (line 458 with details in Appendix), the amount of data used for different methods should be reported directly in the performance results.
> > >
> > > **A8:** We report the amount of clean data used by each method in our experiments in the table below:
> > > | Method Category | Method | SST-2 Clean Data Used | CIFAR-10 Clean Data Used |
> > > | --- | --- | --- | --- |
> > > | Training-time defense | Z-Def | all training data | N/A |
> > > | | CutMix | N/A | all training data |
> > > | Inference-time filtering | ONION | 0 | N/A |
> > > | | ShrinkPad | N/A | 0 |
> > > | Multi-model merging | WAG | 0 | 0 |
> > > | | TIES | 0 | 0 |
> > > | | DARE | 872 | 10,000 |
> > > | | MSD (ours) | 20-50 per class | 20-50 per class |
> > >
> > > Different defenses operate under distinct data assumptions:
> > > * **Full Training Data:** Methods such as Z-Def and CutMix typically rely on the entire clean training set.
> > > * **Zero Clean Data:** Multi-model merging approaches (WAG, TIES) and inference-time filtering methods (ONION, ShrinkPad) do not require clean data.
> > > * **Validation Data:** DARE makes use of all available clean validation samples during its search procedure.
> > >
> > > MSD is designed around the practical assumption that defenders may have access to a small clean validation set. Our empirical analysis shows that 20-50 samples per class are sufficient for MSD to perform effective candidate selection, enabling strong defense performance without dependence on large clean datasets.

---

> > ### Comment · Reviewer_W5FL · 2025-11-21
> >
> > I don't understand the argument based on what previous papers do similarly.
> > If there is a post-training mitigation method that can act on a single model without knowing whether it's poisoned or not
> > (i.e., weaker assumptions),
> > and it does so effectively with lower computation than the proposed model merging method, then why does the fact
> > that there are previously published model merging-methods matter?
> > The potential for this scenario implies that comparisons need to be made with state-of-the-art
> > post training methods that are not model merging with comparisons based
> > on performance, complexity and assumptions highlighted.
> > Moreover, I personally feel that such content in the main body is
> > far more important than the provided analysis of a two-layer neural network,
> > notwithstanding prior papers that analyze such models for their own purposes.

---

> > > ### Author Response · Authors · 2025-11-23
> > >
> > > Thank you for the prompt reply. Here are our responses to the additional questions.
> > >
> > > > **Q9:** The position of model-merging defenses, comparing other  single model post-training backdoor mitigation methods, such as MM-BD [1] and I-BAU [2] is unclear.
> > >
> > > **A9:** We would like to clarify this from three perspectives:
> > >
> > > **1. Two series of defense solutions are orthogonal and complementary to each other.**
> > >
> > > When deploying models, defenders often adopt **multi-layer** defense strategies in which multiple (orthogonal) methods could be considered to improve security. For example, these approaches can be integrated as follow:
> > >
> > > ```
> > >    [Option A: Detection First]                 [Option B: Repair First]
> > >
> > >         MM-BD (Detection)                      Module Switching (Repair)
> > >                │                                         │
> > >                ▼                                         ▼
> > >     Module Switching (Repair)                   MM-BD (Verification)
> > > ```
> > >
> > > Because these methods operate on different mechanisms, stacking them leads to stronger overall robustness. As another example in DAM [4], model merging and unlearning-based defense methods (e.g., AWM [3]) are complementary in and they could form a stronger two-stage defense framework.
> > >
> > > **2. MSD is applicable across broader modalities and architectures.**
> > >
> > > Both [1] and [2] mainly operate in the **vision** domain (and are only evaluated on CNN-based models), whereas our method works for **both text and vision**, and is validated on **Transformer** (both text and ViT models, see Section 5.2), **CNN** architectures (please refer to our response A2 to reviewer 9m3G) and MLP (two-layer network discussed in Section 3). The transfer from Transformer to ViT is smooth, demonstrating the generalization capability of our approach across modalities.
> > >
> > > This is because our method does not rely on modality-specific signals to detect poisoned samples or neurons. Instead, it follows a more general intuition: backdoor attacks, whether in text or vision, build **spurious-correlation shortcuts** within the model parameter space. Disrupting these shortcuts can effectively cut the backdoor transmission.
> > >
> > > Notably, the research trend is moving toward unifying more modalities and tasks within a single model, highlighting the value of more generalizable approaches.
> > >
> > > **3. Potential instability of recommended methods.**
> > >
> > > As reported in another following work [3] (see Table 1), the recommended I-BAU [2] has documented constraints, such as potential utility degradation and sensitivity to the amount of available clean data. Specifically,  I-BAU achieves low ASR when **5000** (much larger than our requirement) clean samples are available, but ASRs of BadNets attack remain **above 90%** when only **50 to 100** samples are available, and the clean accuracy degrades by about **14%** under such settings.
> > >
> > > This aligns with our observations on other existing defenses such as Z-Def ([5], see Table 1 of our paper) in the text domain, which achieves ASR lower than 5% on simple attacks like BadNet but retains 97.3% ASR on more stealthy and advanced attacks like LWS.
> > >
> > > While each method has its strengths, each also has limitations that make it effective in some settings but less so in others. This also underscores the need for a muti-layer defense.
> > >
> > > ---
> > >
> > > Based on the above three reasons and rationalities, we consider methods like MM-BD and I-BAU as complementary defenses rather than direct competitors.
> > >
> > > [1] MM-BD: Post-Training Detection of Backdoor Attacks with Arbitrary Backdoor Pattern Types Using a Maximum Margin Statistic. (IEEE S&P 2024)
> > > [2] Adversarial Unlearning of Backdoors via Implicit Hypergradient. (ICLR 2022)
> > > [3] One-shot Neural Backdoor Erasing via Adversarial Weight Masking. (NeurIPS 2022)
> > > [4] Mitigating the backdoor effect for multi-task model merging via safety-aware subspace. (ICLR 2025)
> > > [5] Mitigating Backdoor Poisoning Attacks through the Lens of Spurious Correlation. (EMNLP 2023)

---

### Official Review · Reviewer_rsY2 · 2025-10-25

**Soundness:** 3
**Presentation:** 3
**Contribution:** 3
**Rating:** 6
**Confidence:** 4

**Summary:**

This paper proposes a module-switching defense (MSD) to mitigate the threat of backdoor attacks. MSD disrupts the backdoor shortcuts formed by backdoor attacks by strategically switching weight modules between multiple models. Compared to the previous weight averaging (WAG) method, MSD achieves stronger defense with fewer models, and additionally exhibits robust defense against collusive attacks.

**Strengths:**

1，	The experiments cover both text and vision tasks and evaluate multiple model architectures.
2，	Section 3 provides a mathematical explanation, justifying why MSD is superior to the weight averaging method.
3，	The authors designed five heuristic rules to guide the module-switching strategies, and the results demonstrate their effectiveness, as removing any of these rules generally leads to performance degradation.
4，	Although the search for module-switching strategies requires an offline cost, the authors report an acceptable runtime, taking 2.6 hours to merge two models and 4.3 hours to merge four models.

**Weaknesses:**

1，	The defensive baselines used for comparison in the vision experiments are outdated. The evaluation should include comparisons against more advanced backdoor defense methods, such as [1] [2] [3].
2，	The experiments for MSD are entirely based on the Transformer architecture; it is unclear whether it can be applied to traditional CNN architectures.
3，	The primary purpose of Adaptive-Patch is to evade activation-based defenses. Hence, strictly speaking, it cannot be considered as an adaptive attack against the MSD method; it is recommended that the authors clarify its role in the discussion of adaptive attacks.
4，	What is the structural overlap between multiple optimal strategies found by different random seeds? Specifically, are there certain layers or modules for which the same source model is consistently selected across these strategies?

[1] Gao Y, Chen H, Sun P, et al. Energy-based backdoor defense without task-specific samples and model retraining[C]//Proc. of International Conference on Machine Learning. 2024: 1-11.
[2] Hou L, Feng R, Hua Z, et al. IBD-PSC: input-level backdoor detection via parameter-oriented scaling consistency[C]//Proceedings of the 41st International Conference on Machine Learning. 2024: 18992-19022.
[3] Xie T, Qi X, He P, et al. Badexpert: Extracting backdoor functionality for accurate backdoor input detection[J]. arXiv preprint arXiv:2308.12439, 2023.

**Questions:**

1，	Can MSD be effective on CNN architectures? If not, what are the reasons?
2，	Similar to WAG, does the MSD method also require that the models intended for merging must be homologous (i.e., possess identical base architectures) and target the same task?
3，	How does the time spent to search for module-switching strategies compare to the key baselines mentioned in the paper (TIES, DARE, and WAG)?

---

> ### Author Response · Authors · 2025-11-21
> **Response to Reviewer rsY2 (1/2)**
>
> We thank the reviewer for their thoughtful feedback and for recognizing the evaluation breadth, the mathematical explanation, the effectiveness of our heuristic design, and the practicality of the reported runtime. Below, we address all points raised.
>
> ---
>
> > **W1:** The defensive baselines used for comparison in the vision experiments are outdated, and the evaluation should include comparisons against more advanced backdoor defense methods, such as [1] [2] [3].
>
> **A1:** We thank the reviewer for the suggestion and have carefully examined the three recommended methods, [1] EBBA+, [2] IBD-PSC, and [3] BaDExpert. However, we consider them to operate from an **orthogonal and complementary angle**, rather than being suitable as direct baselines for our setting.
>
> Approaches [1]-[3] focus on data-level detection or filtering, whereas our work studies how to combine multiple available models to disrupt backdoors at the model level. These two defense dimensions (data-level and model-level) are orthogonal. Accordingly, we compare against baselines that operate under the same multi-model setting, such as WAG, TIES, and DARE. In this sense, methods [1]-[3] are conceptually complementary to MSD rather than competing baselines.
>
> We will cite these works in the related work section and clarify: *"Studies working on poisoned models and sample detection address a complementary defense dimension (data-level filtering), whereas MSD focuses on model-level purification."*
>
> ---
>
> > **W2 & Q1:** The experiments for MSD are entirely based on the Transformer architecture, making it unclear whether MSD can be effective on traditional CNN architectures; whether MSD can be applied to CNNs and, if not, what the reasons would be.
>
> **A2:** Our initial choice to evaluate MSD on Transformer-based models follows from their strong performance across both text and vision tasks. When examining cross-domain applicability, we therefore maintained architectural consistency by extending the evaluation from text Transformers to Vision Transformers.
>
> To further assess the generality of MSD beyond Transformer families, we also extended MSD to CNN-based models during the intense rebuttal period, including ResNet-18 and ResNet-50 on CIFAR-10. The results are shown below:
> | Models | Combination | Method | CACC | Atk1 ASR | Atk2 ASR | AVG. ASR |
> | --- | --- | --- | --- | --- | --- | --- |
> | ResNet18 | BadNet + BATT | no-defense | 95.93 | 98.34 | 99.84 | 99.09 |
> | | | WAG | 96.34 | 10.21 | 13.87 | 12.04 |
> | | | Ours | 94.46 | **10.18** | **13.37** | **11.78** |
> | ResNet18 | BadNet + WaNet | no-defense | 95.79 | 98.34 | 100.0 | 99.17 |
> | | | WAG | 96.14 | **10.94** | 10.15 | **10.55** |
> | | | Ours | 94.41 | 11.26 | **9.91** | 10.59 |
> | ResNet50 | BadNet + BATT | no-defense | 95.13 | 98.42 | 99.81 | 99.12 |
> | | | WAG | 96.61 | 10.29 | 13.71 | 12.00 |
> | | | Ours | 95.59 | **10.02** | **12.11** | **11.07** |
> | ResNet50 | BadNet + WaNet | no-defense | 94.99 | 98.42 | 99.91 | 99.17 |
> | | | WAG | 96.53 | 10.31 | 9.99 | 10.15 |
> | | | Ours | 96.13 | **10.04** | **9.75** | **9.90** |
>
> Settings for adapting MSD to CNNs:
> * We use PyTorch ImageNet-pretrained ResNet models.
> * For MSD search, each BasicBlock is treated as a layer, with its conv-batchnorm pairs considered as modules (ResNet-18: C1 = `conv1+bn1`, C2 = `conv2+bn2`; ResNet-50: C1 = `conv1+bn1`, C2 = `conv2+bn2`, C3 = `conv3+bn3`).
>
> These additional results show that MSD extends naturally to CNN architectures and delivers **comparable or superior robustness** over WAG, supporting its cross-domain applicability beyond Transformer-based models.
>
> ---
>
> > **W3:** The primary purpose of Adaptive-Patch is to evade activation-based defenses; strictly speaking, it cannot be considered an adaptive attack against MSD, and its role should be clarified in the discussion of adaptive attacks.
>
> **A3:** In our evaluation, we include Adaptive-Patch as a challenging backdoor pattern because it learns a complex shortcut in the latent space (“learning a separate shortcut rule” and “fitting a more complicated boundary”), which is directly relevant to our goal of disrupting shortcut pathways within the feedforward structure.
>
> We also emphasize that our evaluation already includes a more realistic MSD-adaptive scenario, in which the attacker could know the deployed select strategy after we published it and retrained only those modules on poisoned data. MSD remains robust under this threat model by generating diverse alternative strategies, as shown in Section 5.4.
>
> To better clarify the role of Adaptive-Patch in our evaluation, we add the following explanation in revision:
>
> * **Line 464:**
>   > We consider two types of threat scenarios: attacks adaptive to MSD and challenging backdoor patterns that introduce stronger shortcut behaviors.
> * **Line 469:**
>   > Second, we evaluate a challenging backdoor pattern, Adaptive-Patch (Qi et al., 2023), which is not MSD-specific but induces more complex shortcut behavior.

---

> > ### Author Response · Authors · 2025-11-21
> > **Response to Reviewer rsY2 (2/2)**
> >
> > > **W4:** What is the structural overlap between multiple optimal strategies found by different random seeds; specifically, whether certain layers or modules consistently select the same source model across strategies.
> >
> > **A4:** Thanks, this is very useful. We performed an overlap analysis across the three strategies presented in Figures 7 and 9 of our paper. The key findings are as follows:
> >
> > 1. **Low structural overlap (6.94%).**
> >    Only 10 out of 144 module positions consistently choose the same source model across all three strategies, indicating minimal agreement among them.
> > 2. **No layer- or module-specific consistency.**
> >    The overlapping positions appear across layers 1, 3, 4, 7, 10, 13, 14, 18, 20, and 24, without concentration in early, middle, or late layers. No specific module type shows higher consistency than others.
> > 3. **High diversity among effective strategies.**
> >    The very low overlap (<7%) suggests that MSD does not rely on modifying particular “critical” layers or modules. Instead, many diverse strategies can sufficiently disrupt shortcut pathways, aligning with our intended design.
> >
> > These observations strengthen MSD’s robustness: the method succeeds by inducing broad structural disruption rather than depending on attack-specific or task-specific critical points, making the resulting strategies transferable and reusable across different scenarios. We will update the manuscript to include this analysis and the corresponding overlap visualization at the end of Section 5.3.
> >
> > ---
> >
> > > **Q2:** Similar to WAG, does MSD also require that the models intended for merging must be homologous (i.e., share identical base architectures) and target the same task?
> >
> > **A5:** Similar to prior work on model merging (WAG, TIES, DARE), our experiments primarily adopt homologous architectures. We follow this controlled setting to evaluate our hypothesis that swapping modules between homologous models, even when both are compromised, can effectively disrupt backdoor transmission.
> >
> > Identical tasks are not required for MSD. We empirically evaluate MSD in a cross-dataset setting where models are trained on related sentiment-analysis tasks but originate from different domains (SST-2 and IMDB). MSD continues to perform well and outperforms WAG, as shown below:
> > | No. | Setting | SST-2 CACC | SST-2 Atk1 ASR | SST-2 Atk2 ASR | AVG. ASR |
> > | --- | --- | --- | --- | --- | --- |
> > | 1 | IMDB-BadNet | 88.6 | 91.8 | - | - |
> > | 2 | SST2-Sent | 96.3 | - | 100.0 | - |
> > | 3 | WAG | 95.5 | 18.3 | 57.2 | 37.8 |
> > | 4 | Ours (MSD) | 95.5 | 19.5 | 33.3 | 26.4 |
> >
> > These results suggest that MSD remains effective in a more attainable and practical scenario.
> >
> > We consider quantifying the minimal architectural and task similarity required in model-merging settings a promising direction for future work.
> >
> > ---
> >
> > > **Q3:** How does the time spent to search for module-switching strategies compare to the key baselines (TIES, DARE, and WAG)?
> >
> > **A6:** We present a time comparison between MSD and the key baselines in the two-model merging setting:
> >
> > | Phase | DARE | TIES | WAG | MSD |
> > | --- | --- | --- | --- | --- |
> > | Search | 2.5 hrs | – | – | 2.6 hrs |
> > | Merge | – | 1 min | 10 s | 16 s |
> >
> > Note that the search cost of MSD is **architecture-dependent only** and can be performed **offline**. Once a strategy is obtained (or retrieved from a community-shared knowledge base), it can be reused across any models sharing the same structure.
> >
> > This amortizes the search effort and yields a negligible deployment cost (16 s). In contrast, deployment-time search methods such as DARE must rerun a greedy search for every new model pair, requiring about 2.5 hours per deployment in our experiments (which is O(n), and our strategy is O(1)). Improving search efficiency remains a promising direction.

---

### Official Review · Reviewer_qum2 · 2025-10-28

**Soundness:** 4
**Presentation:** 3
**Contribution:** 3
**Rating:** 6
**Confidence:** 3

**Summary:**

The paper introduces Module-Switching Defense (MSD), a post-training defense mechanism against backdoor attacks in neural networks. Instead of relying on clean datasets or trusted models, MSD disrupts backdoor “shortcuts” by selectively swapping network modules between potentially compromised models. The authors justify the approach theoretically on two-layer networks and extend it to Transformers using heuristic scoring and an evolutionary search algorithm. Experiments on both NLP and vision datasets demonstrate superior robustness and generalization compared to weight-averaging and other baselines.

**Strengths:**

- The idea of module-level switching as a defense mechanism is novel and distinct from existing model-merging approaches (e.g., WAG, DARE). It offers a new perspective on exploiting architectural modularity for security.
- The theoretical justification in Section 3 is well-motivated, proving stronger backdoor divergence than weight averaging. The empirical analysis aligns with the theory.
- The approach generalizes across modalities (text and vision) and architectures (RoBERTa, BERT, ViT), showing strong versatility.
- MSD operates in post-training settings with limited clean data and no need for retraining, making it suitable for real-world scenarios.

**Weaknesses:**

- The long theory part in Section 3 is not easy to follow. It would be better to add some concise explanations to connect those definitions and the theorem logically.
- The evolutionary search (2M generations, ~2–4 hours) is computationally expensive and may limit deployment practicality; no efficiency analysis or early stopping heuristic evaluation in realistic resource settings is provided.
- Despite the complexity of the method, the improvement in some cases seems to be limited (e.g., Tables 8 and 9), limiting the application of the method.

**Questions:**

- How sensitive is MSD to the choice of heuristic weights? Is there an adaptive mechanism for tuning them automatically?
- Could the evolutionary search be partially amortized or transferred across architectures to further reduce cost?
- How would MSD behave if all source models share identical backdoors in both pattern and location (beyond Table 20 scenarios)?

---

> ### Author Response · Authors · 2025-11-21
> **Response to Reviewer qum2 (1/2)**
>
> We thank the reviewer for their thoughtful feedback and for recognizing the novelty, the strength of our theoretical analysis, and the versatility of MSD across modalities and architectures. Below, we address all points raised.
>
> ---
>
> > **W1:** The long theory part in Section 3 is not easy to follow; it would be better to add concise explanations to connect the definitions and the theorem logically.
>
> **A1:** To improve the readability of Section 3, we plan to clarify it from two aspects:
>
> 1. **[Restructure Section 3]**
>
>    * Lines 133–143: Sec 3.1 Preliminary setup
>    * Lines 144–189: Sec 3.2 Theoretical analysis
>    * Lines 190–215: Sec 3.3 Empirical analysis
>
> 2. **[Guiding sentences for the definitions and the theorems]**
>
>    * Before Definition 1 (Line 144):
>
>      > “We first define the weight averaged (WAG) model and the module switched models, together with the notion of output distances between these combined models and their constituent models. These distances will be used to quantify how WAG and the switched models differ from the constituent backdoor models.”
>
>    * Before Theorem 1 (Line 168):
>
>      > “To show the improved divergence achieved by module switching, we next compare how far the switched models move relative to the constituent backdoor models, in contrast to WAG.”
>
> We will incorporate these revisions into the manuscript. We hope they help make Section 3 clearer, and we are happy to address any further suggestions.
>
> ---
>
> > **W2 & Q2:** The evolutionary search (2M generations, ~2-4 hours) is computationally expensive and may limit deployment practicality; no efficiency analysis or early stopping heuristics are evaluated, and whether the search can be partially amortized or transferred across architectures to further reduce cost.
>
> **A2:** We would like to clarify that the evolutionary search is **structure-driven** rather than *task- or model-dependent*, which allows it to be run entirely offline and makes its cost highly amortizable in practice. Specifically:
>
> 1. **Search can be fully precomputed without access to any real model.**
>    The search only requires a high-level architectural specification (e.g., number of layers and modules per layer). Once the defender determines the model type to be used, the search can be performed offline in advance without querying or loading any actual model, eliminating deployment-time overhead.
> 2. **The discovered strategies transfer across models sharing the same structure.**
>    We empirically show that a single strategy generalizes across tasks, modalities, and checkpoints-for example, across RoBERTa/BERT/DeBERTa and ViT variants-despite their different learned knowledge. This enables the one-time cost to be amortized across many deployments.
> 3. **With a presearched (or community-shared) strategy, the merge cost is negligible.**
>    Using a stored strategy, merging takes only seconds (e.g., 16s for two RoBERTa models), comparable to WAG (10s). This contrasts sharply with deployment-time search methods such as DARE, which require rerunning a greedy search for every deployment and took 2.5 hours to merge two RoBERTa models in our experiments.
> 4. **Early stopping was explored but currently degrades performance.**
>    As discussed in line 437, stopping when no strategy changes after 100K iterations leads to poor accuracy, so we retain the full 2M-step search. We agree that more effective early-stopping criteria are a promising engineering direction.
>
> Overall, because the search is **offline, architecture-only, transferable, and reusable**, its computational cost is effectively amortized and does not hinder deployment practicality.
>
> ---
>
> > **W3:** The improvement in some cases seems limited (e.g., Tables 8 and 9), constraining the applicability of the approach.
>
> **A3:** We acknowledge that performance gains can vary across settings due to differences in utility tasks and backdoor types. In more idealized cases, such as merging a backdoored model with a benign one, the headroom for improvement is naturally limited. In contrast, in the more challenging and realistic scenarios, where both models are backdoored, MSD provides substantial gains.
>
> To offer a clearer overall assessment, we aggregate the ASR reduction ratios relative to the baseline methods across Tables 6-9 for the two-backdoor-model setting:
> | | vs WAG | vs TIES | vs DARE | Avg. |
> | --- | --- | --- | --- | --- |
> | MSD avg. ASR reduction | 17.34% | 37.71% | 31.74% | 28.93% |
>
> These aggregated results show that MSD achieves consistent and robust improvements. Even when gains are modest in some cases, the overall pattern indicates that our approach provides a broadly applicable, general-purpose solution.

---

> > ### Author Response · Authors · 2025-11-21
> > **Response to Reviewer qum2 (2/2)**
> >
> > > **Q1:** How sensitive is MSD to the choice of heuristic weights, and is there an adaptive mechanism for tuning them automatically?
> >
> > **A4:** As described in Section 4.3 (Line 274), we use equal weights $\lambda_k = 1.0$ for all heuristic rules, and this default setting already yields strong defensive performance. Our ablation study in Section 5.3 (Line 449) removes each heuristic rule individually, which is equivalent to setting its $\lambda_k = 0$. The resulting degradation confirms the contribution of all three rules and supports the equal-weight design.
> >
> > We also note that tuning these weights is inherently difficult in our setting because the defender does not have access to any backdoored samples or ASR-related signals. Without such information, standard hyperparameter-tuning procedures cannot be applied.
> >
> > Developing an adaptive mechanism that automatically adjusts heuristic weights using only clean data and status-unknown models is indeed an important research direction, and we believe it merits a dedicated study in future work.
> >
> > ---
> >
> > > **Q2:** How would MSD behave if all source models share identical backdoors in both pattern and location (beyond the scenarios in Table 20)?
> >
> > **A5:** We frame this scenario as a collusive setting in which multiple models share identical backdoors, and note that this case is largely under-explored in model-merging backdoor defense research.
> >
> > Beyond Table 20, which evaluates models with the same backdoor type but different target labels, we also analyze this collusive scenario in Section 5.2 (Line 429). As shown in Table 12, even when some models share fully identical backdoor knowledge, MSD still provides meaningful defense, while WAG degrades noticeably and behaves similarly to a setting with fewer models.
> >
> > For the extreme case where all models share identical backdoors in both pattern and location, we acknowledge that this is an even more challenging setting and is rarely examined in prior work. When every model encodes the same backdoor, the backdoor effectively becomes shared knowledge and is inherently difficult to mitigate, much like how misinformation becomes indistinguishable from truth when all sources repeat the same false narrative, a situation similar of The Truman Show. We consider addressing this fully collusive case as a promising direction for future work.

---

### Official Review · Reviewer_9m3G · 2025-10-30

**Soundness:** 3
**Presentation:** 3
**Contribution:** 3
**Rating:** 4
**Confidence:** 2

**Summary:**

This paper addresses the vulnerability of DNNs to backdoor attacks. The proposed defense MSD disrupts the spurious backdoor shortcuts by selectively swapping modules (such as attention and FFN components) between different models. To optimize switching strategies, MSD uses an evolutionary algorithm under a set of heuristics designed to break potential backdoor propagation paths.

The defense is validated with rigorous experiments under NLP and image classification tasks, using various Transformer models and several datasets.

**Strengths:**

- The idea is intuitive and novel.

- Insightful theoretical motivation and clear visualization (Fig. 2 and Fig. 3)

- The implementation of evolutionary search is practical and methodologically interesting approach.

**Weaknesses:**

- The framework only works when multiple models (possibly all compromised) are available and cannot defend a single compromised model. While this is somewhat fine in the experimental setup, its limited applicability to real-world cases.

- The method is evaluated only on classification tasks with Transformer-based architectures (text and vision). This might be outdated in current DNN research community. The effectiveness of MSD on generative models or non-Transformer architectures (such as CNNs) or other tasks (Object detection, multiple modalities) is not established.

**Questions:**

- How could the framework be adapted or extended to handle scenarios where only a single potentially compromised model is available, as is common in real-world deployments?

- Have you considered evaluating MSD on more diverse architectures or tasks, such as generative models, CNNs, or multimodal systems (LVLMs) to demonstrate broader applicability?

---

> ### Author Response · Authors · 2025-11-21
> **Response to Reviewer 9m3G (1/2)**
>
> We thank the reviewer for their insightful feedback and for recognizing the novelty, clear theoretical motivation, and practical and interesting methodology of our approach. Below, we address all points raised.
>
> ---
>
> > **W1 & Q1:** The framework only works when multiple models (possibly all compromised) are available and cannot defend a single compromised model, limiting applicability to real-world cases; how the framework could be adapted or extended to scenarios where only a single potentially compromised model is available.
>
> **A1:** We would like to clarify this concern from two aspects: (1) the practicality of multiple-model availability, and (2) the scope of our work and potential extensions.
>
> First, as we discuss in **lines 035-043**, our work addresses a growing and practical trend in scenarios where multiple models are available. For example, the Hugging Face Hub hosts over 100 `roberta-large` models fine-tuned for sentiment analysis (e.g., 82 tagged `sentiment`, 28 `sst2`, 13 `imdb`). In the vision domain, similar statistics are observed: the Hub hosts *ViT-based* CIFAR-10 models and *ResNet-50-based* CIFAR-10 models. These statistics show that obtaining multiple compatible models for merging is a realistic assumption across both text and vision applications.
>
> Second, as discussed in **lines 052-057**, our work is situated within the context of utilizing the multiple-model availability property of the model-merging research field. This research line, including our baseline WAG, inherently operates on multiple source models. Our contribution is to provide a robust defense within this established paradigm. We improve its practicality by relaxing the assumption that any model must be benign and by exploring methods to reduce the number of required models to **2**, a configuration that is entirely feasible in real-world deployments, as evidenced above.
>
> Since merging multiple models ultimately yields a single unified model, defenses that target a single compromised model are largely orthogonal to our work. We contend that a robust security paradigm should apply multi-layer defenses, such as MSD (or WAG) prior to deployment, complemented by single-model defenses during inference.

---

> > ### Author Response · Authors · 2025-11-21
> > **Response to Reviewer 9m3G (2/2)**
> >
> > > **W2 & Q2:** The method is evaluated only on classification tasks with Transformer-based architectures (text and vision), which may be outdated; whether MSD can be evaluated on generative models, CNNs, multimodal systems (e.g., LVLMs), or other tasks such as object detection to demonstrate broader applicability.
> >
> > **A2:** We would like to clarify this concern from two aspects: (1) the relevance of our studied tasks, and (2) additional experiments on CNN-based models.
> >
> > First, we choose the tasks and models primarily based on the experimental setup of the baseline method WAG, to ensure a fair and controlled comparison. While WAG focuses on the text domain, we extend the evaluation to the vision domain to examine cross-domain generalizability (line 335). Classification tasks based on Transformer architectures remain widely used and actively studied in contemporary backdoor research across both text and vision ([A]-[G]), reaffirming the relevance of our chosen setting.
> >
> > Second, to further explore the generalization of our method during the intense rebuttal period, we apply MSD to CNN-based models, including ResNet-18 and ResNet-50. The results on CIFAR-10 are shown below:
> > | Models | Combination | Method | CACC | Atk1 ASR | Atk2 ASR | AVG. ASR |
> > | --- | --- | --- | --- | --- | --- | --- |
> > | Resnet18 | BadNet + BATT | no-defense | 95.93 | 98.34 | 99.84 | 99.09 |
> > | | | WAG | 96.34 | 10.21 | 13.87 | 12.04 |
> > | | | Ours | 94.46 | **10.18** | **13.37** | **11.78** |
> > | Resnet18 | BadNet + WaNet | no-defense | 95.79 | 98.34 | 100.0 | 99.17 |
> > | | | WAG | 96.14 | **10.94** | 10.15 | **10.55** |
> > | | | Ours | 94.41 | 11.26 | **9.91** | 10.59 |
> > | Resnet50 | BadNet + BATT | no-defense | 95.13 | 98.42 | 99.81 | 99.12 |
> > | | | WAG | 96.61 | 10.29 | 13.71 | 12.00 |
> > | | | Ours | 95.59 | **10.02** | **12.11** | **11.07** |
> > | Resnet50 | BadNet + WaNet | no-defense | 94.99 | 98.42 | 99.91 | 99.17 |
> > | | | WAG | 96.53 | 10.31 | 9.99 | 10.15 |
> > | | | Ours | 96.13 | **10.04** | **9.75** | **9.90** |
> >
> > We outline the settings used for adapting MSD to CNN models:
> > * We use PyTorch ImageNet-pretrained ResNet models.
> > * For MSD search, each BasicBlock is treated as a layer, with its conv–batchnorm pairs considered as modules (ResNet-18: C1 = `conv1+bn1`, C2 = `conv2+bn2`; ResNet-50: C1 = `conv1+bn1`, C2 = `conv2+bn2`, C3 = `conv3+bn3`).
> >
> > These additional results confirm that MSD’s applicability extends beyond Transformer models. We respectfully emphasize that our primary contribution is the theoretical analysis of disrupting backdoor-induced spurious correlations via module switching, which we demonstrate in representative and widely studied settings. While other modalities (e.g., generative models, multimodal systems, object detection) are indeed of interest, exploring all of them is beyond the scope of a single 9-page paper, and we agree they represent valuable future directions.
> >
> > [A] Arora et al. Here's a Free Lunch: Sanitizing Backdoored Models with Model Merge. (ACL 2024)
> > [B] Yi et al. BadActs: A Universal Backdoor Defense in the Activation Space. (ACL 2024)
> > [C] Chen et al. PKAD: Pretrained Knowledge Is All You Need to Detect and Mitigate Textual Backdoor Attacks. (EMNLP 2024)
> > [D] Zhang et al. BadWindtunnel: Defending Backdoor in High-Noise Simulated Training with Confidence Variance. (ACL 2025)
> > [E] Zhao et al. Defense Against Backdoor Attack on Pre-trained Language Models via Head Pruning and Attention Normalization. (ICML 2024)
> > [F] Yang et al. Mitigating the Backdoor Effect for Multi-Task Model Merging via Safety-Aware Subspace. (ICLR 2025)
> > [G] Hu et al. A Closer Look at Backdoor Attacks on CLIP. (ICML 2025)

---

### Official Review · Reviewer_c8Qa · 2025-11-01

**Soundness:** 3
**Presentation:** 3
**Contribution:** 2
**Rating:** 4
**Confidence:** 4

**Summary:**

The paper proposes the Module-Switching Defense (MSD), a post-training technique to disrupt backdoor shortcuts in neural networks. MSD works by selectively exchanging weight modules between multiple trojan models. Theoretically, the analysis on two-layer networks proving module switching achieves higher backdoor divergence than weight averaging (WAG). For Transformer architectures, MSD employs an evolutionary algorithm guided by heuristic rules to find optimal switching strategies.

MSD requires fewer models (as few as 2) compared to WAG (3-6 models), and works without trusted data or knowledge of attack types. Moreover, it has good transferability across models with the same architecture (e.g., RoBERTa, BERT, DeBERTa) and work across both text and vision domains.

**Strengths:**

The paper provides theoretical analysis to justify the module-switching approach. Theorem 1 formally proves that module switching achieves higher backdoor divergence than weight averaging (WAG) under identity activation, while Proposition 1 guarantees the existence of at least one switched model that outperforms WAG.

MSD is resource-efficient, requiring only 2 models, compared to 3-6 models needed by methods like WAG and DAM. The method operates with minimal clean data, and requires no access to poisoned data, trusted reference models, or prior knowledge of attack types.

The method demonstrates exceptional transferability across multiple dimensions. The searched strategies are architecture-agnostic and transfer seamlessly across different models sharing the same structure. Also, MSD maintains effectiveness across diverse domains (text and vision) and multiple datasets.

**Weaknesses:**

1. MSD operates as an incremental refinement of existing model merging techniques like WAG. While the paper successfully demonstrates a theoretical superiority in maximizing backdoor divergence, the core insight that combining models can suppress spurious correlations was already established. The introduction of evolutionary search serves primarily to optimize the composition step rather than introducing a surprising, non-obvious insight. The work therefore lacks the novelty expected of a top-tier contribution.

2. The empirical evaluation relies heavily on datasets and model architectures that are either small or now considered non-state-of-the-art. Specifically, testing is performed on NLP datasets like SST-2 and AG News, and vision datasets like CIFAR-10. While the method is tested on Transformer-based architectures, the explicit validation against modern, large-scale safety-aligned LLMs is missing.

3. The feasibility of the time-consuming evolutionary search does not scale well to the demands of modern LLM fine-tuning and deployment cycles. The paper fails to sufficiently address the scalability and computational cost implications of running an architecture search for models with larger layer counts or internal complexity.

**Questions:**

1. Given the computational bottleneck of the architectural search, the authors should provide an analysis of scalability or a proposed roadmap for a more computationally efficient module selection mechanism to justify the method's viability in large-scale deployment.

2. Does the module selection strategy offer substantial performance improvements over simple yet effective fusion baselines (like TIES/DARE), or is the advantage primarily marginal optimization?

3. The empirical validation relies heavily on relatively older, smaller classification datasets and models (e.g., RoBERTa and BERT). Given the significant architectural disparity between these encoder-only models and the decoder-only generative LLMs that dominate the current threat landscape (e.g., GPT-like models), the authors should enhance the practical relevance of the findings by providing a transferability analysis on a recent generative LLM architecture. To what extent do these results accurately represent the security challenges posed by modern LLMs?

---

> ### Author Response · Authors · 2025-11-21
> **Response to Reviewer c8Qa (1/2)**
>
> We thank the reviewer for their insightful feedback and for acknowledging our contributions and novelty in theoretical analysis, resource-efficient setting, and exceptional strategy transferability. Below, we address all points raised.
>
> ---
>
> > **W1:** MSD operates as an incremental refinement of existing model merging techniques like WAG, the core insight that combining models can suppress spurious correlations was already established.
>
> **A1:** We would like to clarify that the insight that strategically **fusing** modules from different models can suppress spurious correlations is **our core contribution, and a novel concept that is not established by WAG**.
>
> WAG does not inherently suppress spurious correlations. Instead, its averaging mechanism preserves shortcut features present in all source models. This lack of a disruption mechanism explains its significant underperformance in settings with few available models (line 376) and in collusive backdoor scenarios (line 429) in Section 5.2.
>
> Our advantages against WAG in suppressing spurious correlations is supported by comprehensive experiments. This includes WAG's poor performance with two models (Table 1), and also the collusive scenario in Table 12 (where models share identical backdoors). In that scenario, WAG largely preserves the shared shortcut between collusive models and its performance degrades, whereas our MSD approach is still able to strategically combine modules and disrupt the backdoor. Therefore, MSD is not an incremental refinement but a new mechanism based on a distinct theoretical insight.
>
> ---
>
> > **W2 & Q3:** The empirical validation relies heavily on older, smaller datasets and encoder-only models, without evaluation on modern large-scale generative LLMs; the authors should clarify the transferability of the results and their relevance to contemporary LLM security challenges.
>
> **A2:** We would like to clarify the scope and relevance of our validation.
>
> Firstly, our module-switching approach is grounded in a theoretical analysis showing that disrupting backdoor-induced shortcut dependencies mitigates attacks. We empirically validate this insight in representative and widely studied Transformer-based text and vision settings, which continue to serve as standard benchmarks in contemporary backdoor-defense research. During the rebuttal period, we further extended our study to CNN-based architectures, and we report these new results in our response A2 to reviewer 9m3G.
>
> Many real-world applications, such as hate-speech detection, safety filtering, and other high-throughput moderation pipelines, still rely on encoder-based models due to their efficiency in cost and time. Our chosen datasets (sentiment analysis, topic classification, and hate-speech detection) reflect major real-world applications widely used in production systems by social media and content platforms.
>
> While our experiments now span both Transformer and CNN families and cover representative real-world tasks, evaluating MSD on large-scale generative LLMs  is a promising direction for future work.

---

> > ### Author Response · Authors · 2025-11-21
> > **Response to Reviewer c8Qa (2/2)**
> >
> > > **W3 & Q1:** The evolutionary search appears computationally expensive and may not scale to modern LLMs with larger layer counts or internal complexity; the authors should provide an analysis of scalability to justify the method’s viability in large-scale deployment.
> >
> > **A3:** We would like to highlight that the evolutionary search is a **one-time, offline cost per architecture**, not a cost incurred per task or per deployment. This distinction is essential. For example, the unified strategy found in Fig. 6 generalizes well across different model families, multiple tasks, and various poisoning rates.
> >
> > This one-time search runs entirely on CPU (without requiring high-end GPUs) and completes within several hours, which we believe is a reasonable expense for a robust defensive strategy. Once an offline strategy is optimized, the online MSD defense-performing the actual module switching-is extremely efficient, usually taking *less than one minute* for each particular model.
> >
> > Though our primary focus is on providing a **theoretical understanding** of how disrupting spurious correlations mitigates backdoors and **implementing this insight** for defence, we acknowledge that further reducing search overhead is beneficial. Our research community can share such effort by **releasing the discovered strategies as public artifacts**, and we plan to share ours upon acceptance.
> >
> > ---
> >
> > > **Q2:** Does the module selection strategy offer substantial performance improvements over simple yet effective fusion baselines (like TIES/DARE), or is the advantage primarily marginal optimization?
> >
> > **A4:** Yes, the relative performance improvement is significant and stable. We compute the relative ASR reduction ratios of MSD compared to WAG, TIES, and DARE using the results reported in Table 1. The detailed reductions are shown below:
> >
> > | Attack Type | Ours ASR | WAG ASR | Reduction vs WAG | TIES ASR | Reduction vs TIES | DARE ASR | Reduction vs DARE |
> > | --- | --- | --- | --- | --- | --- | --- | --- |
> > | badnet + insertsent | 22.0 | 31.9 | 31.0% | 52.9 | 58.4% | 47.1 | 53.3% |
> > | badnet + hiddenkiller | 34.1 | 46.4 | 26.5% | 63.6 | 46.4% | 41.9 | 18.6% |
> > | badnet + lws | 40.4 | 62.2 | 35.0% | 77.1 | 47.6% | 61.4 | 34.2% |
> > | badnet + benign | 12.2 | 39.3 | 69.0% | 69.2 | 82.4% | 43.2 | 71.8% |
> >
> > These results show that MSD consistently reduces ASR relative to all three baselines, often by a large margin (commonly around 20%-70%). Although different attack combinations exhibit natural variance, the overall trend is stable.
> >
> > Overall, the relative reduction of ASR demonstrates that MSD offers substantial defensive improvement over existing fusion-based approaches.

---

### Author Response · Authors · 2025-12-04
**Summary to ACs and Reviewers**

Dear Area Chairs and Reviewers,

We sincerely thank you for the time and effort devoted to reviewing our submission, as well as for the constructive feedback that has helped further strengthen the paper.

As the discussion phase is coming to a close, we would like to provide a concise summary of the key points raised in the reviews, the clarifications provided in our rebuttal, and the corresponding manuscript revisions.

---

* **Strengths Recognized by Reviewers**

  1. The idea of module-level switching is intuitive and novel (*qum2*, *9m3G*), offering a new security perspective via architectural modularity (*qum2*).
  2. The theoretical justification is insightful (*9m3G*) and well-motivated (*qum2*), providing formal support for module switching (*c8Qa*, *rsY2*).
  3. The method shows strong transferability across modalities, architectures, and tasks (*c8Qa*, *qum2*, *rsY2*).
  4. The method is practical (*9m3G*), resource-efficient (*c8Qa*), and suitable for real-world deployment (*qum2*).

---

* **Rebuttal Clarification and Manuscript Revision**

  1. **Applicability beyond Transformers (e.g., CNNs) (*9m3G*, *rsY2*)**
     * *Response:* We extended validation to ResNet models, confirming cross-modality and cross-architecture generalization (A2 to *9m3G*, *rsY2*).
     * *Revision:* Added discussion on vision tasks and models in Section 4.1 (line 258), Section 4.4 (line 345), and Section 5 (lines 406 and 493).

  2. **Clarity of the theoretical analysis (*qum2*)**
     * *Response:* We restructured Section 3 into three subsections and added several guiding sentences to improve readability, as explained in our response A1 to reviewer *qum2*, and highlighted the changes in the revision $(\text{in } \textcolor{magenta}{magenta})$.
     * *Revision:* Added subsection titles (lines 137, 150 and 206) and guiding sentences (lines 152 and 181).

  3. **Computational cost of evolutionary search (*c8Qa*, *qum2*, *rsY2*, *W5FL*)**
     * *Response:* Our method is computationally efficient, since (1) MSD is task-independent (sharing computation across different tasks); (2) it runs offline (even before the model is trained/finetuned); and (3) its cost is amortized (shared) across models with an identical structure (A6 to *rsY2*).
     * *Revision:* Added efficiency analysis in Section 5.3 (line 522).

  4. **Performance improvements over baselines (*c8Qa*, *qum2*)**
     * *Response:* We reported relative ASR reduction, with average improvements over 20% against competitive baseline methods under two-model merging settings.

  5. **Comparison to certain vision-domain defenses (*rsY2*, *W5FL*)**
     * *Response:* We clarified these methods focus on detection, while our approach addresses model-level repair; the two series of defense methods are complementary.
     * *Revision:* Added references and explanation in Section 2 (line 123).

  6. **Role of Adaptive-Patch as an adaptive attack (*rsY2*)**
     * *Response:* We clarified that Adaptive-Patch serves as a challenging backdoor pattern that induces more complex backdoor shortcuts. The experiments show that MSD also shows advantages in defending against such attacks.
     * *Revision:* Added clarification in Section 5.3 (lines 505 and 510).

  7. **Structural overlap among searched strategies (*rsY2*)**
     * *Response:* Overlap analysis across seeds shows low overlap (<7%), indicating decent structural diversity.
     * *Revision:* Added discussion in Section 5.2 (line 470).

  8. **Some appreciated scenarios to apply our defense**
     * *Response on* **model availability:** We collected and added corresponding statistics (A1 to *9m3G*).
     * *Response on* **task similarity:** We added cross-domain merging results (A5 to *rsY2*).
     * *Response on* **identical-backdoor settings:** We discussed collusive attacks in the original submission and provided further clarification (A5 to *qum2*, A4 to *W5FL*).

  9. **Novelty relative to WAG (*c8Qa*)**
     * *Response:* The core contribution of MSD is to introduce a novel approach that disrupts spurious-correlation shortcuts through **module switching**. It is conceptually distinct from WAG, which mixes the models by averaging the model parameters and all backdoor information is unavoidably inherited. The advantage is also supported by superior performance of our method on experiments using few models and collusive attack scenarios (A1 to *c8Qa*).

---

We believe that the clarifications and additional analyses provided during the rebuttal period help present a clear and complete picture of our contributions. We’ve incorporated all of them into our revision (highlighted in $\textcolor{magenta}{magenta}$), which significantly strengthens this work. We sincerely appreciate the reviewers’ and area chairs’ thoughtful suggestions, careful evaluation and the time they dedicated to assessing our submission.

---

### Meta-Review · Area_Chair_Jay2 · 2026-01-07

**Summary:**

The reviewers initially raised several significant concerns, including limited technical novelty, empirical evaluation being limited, the evolutionary search being computationally expensive, and the improvement in some cases seems to be limited.

After reading the rebuttal, it felt that most questions regarding experimental details and missing experiments have been addressed. While the issue of novelty is not fully resolved, the work clearly possesses merit and contributes meaningfully to the field. Therefore, my overall recommendation is Acceptance. Authors should carefully revise their paper according to the review comments.

**Reviewer Concerns:**

After reading the rebuttal, it felt that most questions regarding experimental details and missing experiments have been addressed.

**Reviewer Scores:**

Reviewer qum2 and rsY2 may maintain their positive scores.
 Reviewer 9m3G and c8Qa may raise the score from 4 to 6.
 Reviewer W5FL may maintain the negative score.

---

### Decision · Program_Chairs · 2026-01-26

Accept (Poster)